# (How) Can Transformers Predict Pseudo-Random Numbers?

**Tao Tao** [* 1]  **Darshil Doshi** [* 1]  **Dayal Singh Kalra** [* 2]  **Tianyu He** [* 1]  **Maissam Barkeshli** [1 3]

{tao2021, ddoshi, dayal, tianyuh, maissam}@umd.edu

## Abstract

Transformers excel at discovering patterns in sequential data, yet their fundamental limitations and learning mechanisms remain crucial topics of investigation. In this paper, we study the ability of Transformers to learn pseudo-random number sequences from linear congruential generators (LCGs), defined by the recurrence relation $x_{t+1} = ax_t + c \mod m$. We find that with sufficient architectural capacity and training data variety, Transformers can perform in-context prediction of LCG sequences with unseen moduli ($m$) and parameters ($a, c$). By analyzing the embedding layers and attention patterns, we uncover how Transformers develop algorithmic structures to learn these sequences in two scenarios of increasing complexity. First, we investigate how Transformers learn LCG sequences with unseen $(a, c)$ but fixed modulus; and demonstrate successful learning up to $m = 2^{32}$. We find that models learn to factorize $m$ and utilize digit-wise number representations to make sequential predictions. In the second, more challenging scenario of unseen moduli, we show that Transformers can generalize to unseen moduli up to $m_{\text{test}} = 2^{16}$. In this case, the model employs a two-step strategy: first estimating the unknown modulus from the context, then utilizing prime factorizations to generate predictions. For this task, we observe a sharp transition in the accuracy at a critical depth $d = 3$. We also find that the number of in-context sequence elements needed to reach high accuracy scales sublinearly with the modulus.[†]

---

[*]Equal contribution – authors listed in pseudo-random order. [1]Department of Physics, University of Maryland, College Park, USA [2]Department of Computer Science, University of Maryland, College Park, USA [3]Joint Quantum Institute, University of Maryland, College Park, USA. Correspondence to: Maissam Barkeshli <maissam@umd.edu>.

*Proceedings of the 42$^{nd}$ International Conference on Machine Learning*, Vancouver, Canada. PMLR 267, 2025. Copyright 2025 by the author(s).

[†]We open source the code to reproduce our results: https://

## 1. Introduction

Transformer-based language models have proven to be extremely powerful sequence generative models. With copious amounts of training data and computational resources, they can identify and learn complex patterns from training corpora, resulting in numerous remarkable capabilities (Vaswani et al., 2017; Dosovitskiy et al., 2021). Recent research has demonstrated that these models, when provided with sufficient context and inference compute, can acquire new patterns and capabilities without additional training through techniques such as in-context learning (Radford et al., 2019) and chain-of-thought reasoning (Wei et al., 2023). While these models have achieved unprecedented success, understanding what underlying patterns are learned and how they learn them remains a significant challenge. Pseudo-Random Number Generators (PRNGs) represent an interesting test case for exploring these challenges. These algorithms, which are fundamental to modern cryptography and computer science, are designed to produce outputs that pass statistical tests for randomness, but nevertheless arise from mathematical patterns that could potentially be learned by sufficiently powerful sequence models.

This intersection between Transformer models' pattern-learning capabilities and the structured yet obfuscated nature of PRNG outputs raises intriguing questions about both the capabilities and limitations of these models. Can Transformers learn to predict PRNG outputs given sufficient training data, model capacity, and context? If so, what implications does this have for our understanding of both Transformer architectures and PRNGs? Do the Transformers learn the underlying generating algorithm or merely detect shortcuts and spurious patterns? What effect do model capacity, data variety, training methodologies, and context length have on the capabilities of Transformers?

This work aims to answer these questions by focusing on learning sequences obtained from linear congruential generators (LCGs) using GPT-style autoregressive Transformers. We demonstrate how Transformers can successfully learn LCGs with moduli up to $m = 2^{32}$. We perform interpretability analyses, uncovering emergent structures in

github.com/dayal-kalra/transformer-prng.git

the embedding layers, attention heads, and underlying algorithms that the Transformer uses to learn the sequences. We also perform several systematic scaling analyses to understand the effect of architecture and sequence complexity on model performance and in-context learning ability.

### 1.1. Related works

Our study on the learnability of PRNGs for Transformers touches on several modern and classic topics.

**Interpretability and Modular Arithmetic:** A growing body of work examines the circuits, algorithms and structures learned by Transformers (Sharkey et al., 2025; Olsson et al., 2022; Ahn et al., 2023; von Oswald et al., 2023; Akyürek et al., 2023; Hendel et al., 2023; Liu et al., 2024). A notably fruitful setting involves simple modular arithmetic problems (Power et al., 2022; Gromov, 2023; Nanda et al., 2023; Zhong et al., 2023; Doshi et al., 2024a;b; He et al., 2024). Our work adds to this by reverse-engineering the underlying algorithms and uncovering emergent structures in learning pseudo-random number sequences.

**Cracking PRNGs:** There is a classic duality between cryptography and learning theory (Rivest, 1991), and cracking PRNGs is an important topic in cryptography. Nevertheless, deep learning-based attacks have received limited attention in the post-Transformer era. Amigo et al. (2021) demonstrated that a fully-connected neural network can predict the outputs of a modified LCG with fixed (irrational) parameters $(a, c, m) = (1, \pi, 1)$. In comparison, we systematically analyze the harder cases of unseen parameters using Transformers, reverse-engineer the learned algorithms, and study effects of scale and complexity.

**Formal Grammars:** LCG can also be viewed as a formal language (Type-3 regular grammar) lying within the Chomsky hierarchy (Chomsky, 1956). Formal languages provide an interesting setting for synthetic datasets that can be used to understand the properties of neural networks in controlled settings (Delétang et al., 2023; Allen-Zhu & Li, 2024; Cagnetta et al., 2024; Cagnetta & Wyart, 2024).

**Chaotic time-series:** A major application of neural networks is predicting time-series for chaotic dynamics, such as weather prediction (Lam et al., 2023) and financial modeling (Lam et al., 2023) and financial modeling. PRNGs provide an analog of such dynamics in the discrete setting.

### 1.2. Linear Congruential Generators

LCG is a simple PRNG that generates the next number in a sequence $(x_0, x_1, \ldots, x_t)$ according to the map:

$$x_{t+1} = (ax_t + c) \mod m, \tag{1}$$

where $m > 0$ is the modulus, $0 < a < m$ is the multiplier and $0 \le c < m$ is referred to as the increment. An LCG map

is uniquely defined by the choice of $m, a, c$ and the initial seed $x_0$. An important quantity that determines the complexity of an LCG sequence is its period: $1 \le \mathcal{T}_m(a, c) \le m$. As we will show in the following sections, the period of a sequence plays a major role in the difficulty of prediction with Transformers. According to the Hull-Dobell Theorem (Hull & Dobell, 1962), the period $\mathcal{T}_m(a, c) = m$ if and only if the values of $a$ and $c$ satisfy the following criteria: (i) $m$ and $c$ are coprime, (ii) $a - 1$ is divisible by all prime factors of $m$, (iii) $a - 1$ is divisible by $4$ if $m$ is divisible by $4$. We evaluate (test) all our models exclusively on sequences that obey the criteria of this theorem.

LCGs are widely utilized for their speed and simplicity, often forming the core of more complex PRNGs like PCG-64, which is used in NumPy. LCGs perform poorly at small bit sizes but improve rapidly with larger state sizes. For instance, an LCG with 88 bits of state can pass the stringent BigCrush randomness test (O'Neill, 2014).

## 2. Training Setup

We train decoder-only Transformers to autoregressively predict the next number in LCG sequences. This means it takes as input an LCG sequence $(x_0, \cdots, x_{L-1})$, outputs a sequence $(y_0, \cdots, y_{L-1})$, and trained so $y_t$ matches $x_{t+1}$.

To predict an unknown LCG sequence, the Transformer needs to infer $m, a$, and $c$ in-context. We test the model's generalization ability in two distinct paradigms of increasing difficulty: **FM:** The model is trained and tested on sequences with a fixed modulus $m$. **UM:** Model is trained on varying moduli, and tested on unseen moduli $M_{\text{test}} = \{m_{\text{test}}\}$. We highlight the key details of our experimental setups here and provide an extensive discussion in Appendix A.

### 2.1. Dataset Generation and Evaluation

The settings below are used in Sections 3 and 4. In order to achieve better performance, we used a larger and higher-quality dataset in Section 5, which we detail later.

**Fixed Modulus (FM):** Given a modulus $m$, we apply the Hull-Dobell Theorem to determine the possible values of $(a, c)$ that maximize the period. We then randomly select $64$ values of $a$ and $c$ to generate the test dataset. To generate the training dataset, we *exlude* these test choices of $(a, c)$ and uniformly sample $N = 100,000$ LCG sequences of length $L + 1$ (where $L$ is the context length), with $n_a$ values of multipliers and $n_c$ values of increments. For each set of parameters $(a, c)$, we sample an LCG sequence with a randomly selected initial seed $x_0$. Note that the training dataset includes sequences with varying periods, while the test data only contains sequences that maximize the period.

**Generalization to Unseen Modulus (UM):** In this more

challenging paradigm, we first select a set of test moduli $M_{\text{test}} = \{m_{\text{test}}\}$ that would be reserved exclusively for evaluation. For each test modulus $m_{\text{test}} \in M_{\text{test}}$, we determine the values of $(a, c)$ that maximize the period. We then randomly select $64$ values of $a$ and $c$ each to generate the test dataset. These $64^2$ $(a, c)$ pairs are not considered while generating the training dataset.

For the training dataset generation, we sample $n_m$ modulus values from the range $[L, m_{\text{max}}]$, with $m_{\text{max}} = \lfloor 1.2 \max(M_{\text{test}}) \rfloor$, while excluding all the values in $M_{\text{test}}$. For each modulus value $m$, we uniformly select $n_a$ multipliers and $n_c$ increments, excluding the ones reserved for testing. For each triplet $(a, c, m)$, we generate a sequence of length $L + 1$ using a randomly selected initial seed $x_0$. This results in a total of $N = n_m \times n_a \times n_c$ training sequences.

We found that $n_m \gtrsim m_{\text{test}}/4$ yields good generalization performance. Based on this relationship and our target total number of training examples $N$, we sample $n_a = n_c = \sqrt{\frac{N}{m_{\text{test}}/4}}$ values of multipliers and increments. Unless explicitly specified, we use this setting as the default configuration for all experiments.

In both paradigms, test accuracy is averaged over all $(a, c)$ test pairs and multiple initial seeds $x_0$ per pair. Accuracies are tracked at all sequence positions $1 \leq t \leq L$.

### 2.2. Tokenization, Architecture, and Optimizer

In Sections 3 and 4, each number is tokenized as a unique token, using a dictionary of size $m$ (**FM**) or $m_{\text{max}}$ (**UM**). We employ GPT-style Transformers with learnable positional embeddings and weight tying (Press & Wolf, 2017).

When we scale up to larger moduli in Section 5, we restrict the dictionary size to $b$ by tokenizing each number in the sequence in base-$b$ (e.g. $b = 2^8, 3^5$). This results in $\lceil \log_b m \rceil$ tokens for each number. We also apply (modified) abacus positional embeddings (McLeish et al., 2024).

The model architecture is characterized by the number of blocks (depth), embedding dimension ($d_{\text{model}}$), number of attention heads ($n_{\text{heads}}$). Models are trained with AdamW (Loshchilov & Hutter, 2019) and CrossEntropy loss.

## 3. Training Results

We begin by investigating the minimal model that can solve the two tasks in consideration. Surprisingly, we found that Transformers only require one layer and one attention head to learn the **FM** task, as shown in Figure 2 (a) (for further results, see Appendix B). Conversely, the **UM** task requires a stronger architecture and careful hyperparameter tuning. Appendix C shows that model performance depends on the modulus, with prime moduli being challenging in the **FM**

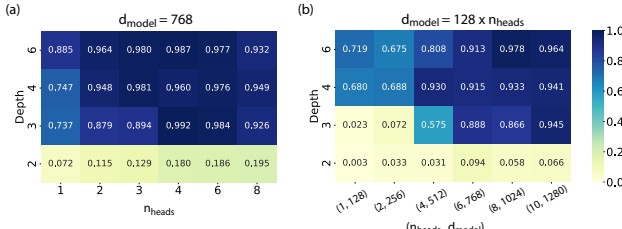

*Figure 1.* Accuracy of predicting the last number (token) in the sequence: phase diagrams w.r.t. various depths and $n_{\text{heads}}$ values. (a) $m_{\text{test}} = 2048$, constant width: $d_{\text{model}} = 768$. (b) $m_{\text{test}} = 4096$, width scaled proportionally: $d_{\text{model}} = 128 \times n_{\text{heads}}$.

setting but not in the **UM** setting.

In Figure 1 we show how the performance varies with model depth and the number of attention heads. In Figure 1(a) we keep the embedding dimension fixed to $d_{\text{model}} = 768$, whereas in Figure 1(b) we scale it proportionally to the number of heads ($d_{\text{model}} = 128 \times n_{\text{heads}}$.). In both cases, we find that a minimum of three layers are required for effective generalization, with performance degrading sharply below this threshold. Further analysis across multiple $m_{\text{test}}$ values (see Appendix D) confirms that this minimal depth requirement is universal. We also observe that additional attention heads improve model performance, with substantial gain occurring when increasing from one to two heads.

Several prior studies have also observed sharp changes in model capabilities as a function of model depth. This includes induction head formation (Olsson et al., 2022), in-context learning of modular addition (He et al., 2024) and various in-context generalization tasks (Chen & Zou, 2024). In general it is unclear to what extent these sharp depth-dependences are due to jumps in expressivity or trainability.

The **UM** task shows strong sensitivity to hyperparameters. As we increase $m_{\text{test}}$ while keeping the model size fixed, we observe two key phenomena: the optimal learning rate ($\eta$) and weight decay strength ($\lambda$) shift significantly, and simultaneously, the range of hyperparameter resulting in effective performance narrows (see Appendix A.5).

We then carefully examine the training dynamics in both **FM** and **UM** settings (Figure 2). We categorize the training sequences into two groups: i) sequences with periods shorter than the context length, which can be solved through simple copying, and ii) sequences with periods longer than the context length, which require the model to deduce underlying rules for prediction. Our analysis reveals that the model first acquires copying ability for group i) in the early stages of training, and later "groks" the solution for group ii) (Power et al., 2022). Notably, the model's ability to generalize to test modulus $m_{\text{test}}$ emerges simultaneously with this grokking phenomenon. These results demonstrate that

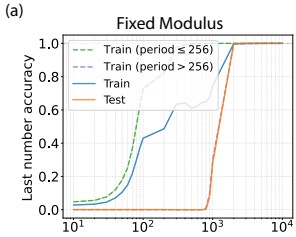
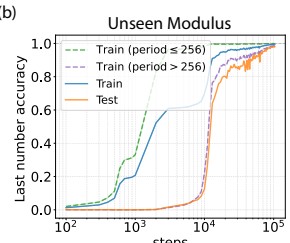

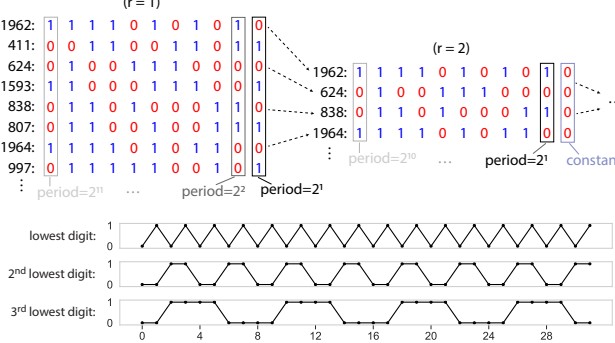

*Figure 2.* Training/test accuracy curves for predicting last number (token). (a) **FM:** ($m = 2048, \text{depth} = 1, n_{\text{heads}} = 1, d_{\text{model}} = 768$) Test accuracy "groks" when training accuracy reaches near-100%. (b) **UM:** ($m_{\text{test}=2048}, \text{depth} = 6, n_{\text{heads}} = 4, d_{\text{model}} = 768$) Test accuracy "groks" simultaneously with training accuracy on sequences with period longer than context length ($\mathcal{T}_m > L = 256$), indicating delayed discovery of underlying rules.

the model develops different capabilities at distinct stages of training, with generalization ability emerging only after the model learns the underlying rules through solving the more challenging sequences. In Appendix E, we present an ablation study where models are trained exclusively on either short-period or long-period sequences. Our findings indicate that training exclusively on long-period sequences enables model generalization.

# 4. Interpreting How Transformers Predict PRNGs

In this section, we uncover the underlying algorithms implemented by the models for both **FM** and **UM** cases. While certain details of the algorithms differ in the two cases, they share common properties originating from the underlying LCG structure. We first discuss properties of LCG sequences that will be useful in interpreting model behaviors.

## 4.1. Residual Number System Representations

Consider an LCG sequence with modulus $m = 2048 = 2^{11}$. Each number in this sequence can be represented as an 11-digit binary number:

$$x \bmod 2^{11} = \alpha_0 \, 2^0 + \alpha_1 \, 2^1 + \cdots + \alpha_{10} \, 2^{10} , \quad (2)$$

where $\{\alpha_0, \ldots, \alpha_{10}\}$ are the binary-valued digits (bits).

A useful property of LCGs with modulus $m = 2^{(\cdot)}$ is that each digit in the binary representation has a fixed period along the sequence. As shown in Figure 3, for a sequence of period $\mathcal{T}_m = m = 2^{11}$, the $w^{th}$ lowest digit has a period of $2^w$ (Knuth, 1997). Thus, lower (higher) digits have smaller (larger) periods along LCG sequences. (See Appendix F.1 for a detailed derivation.) We will see later that trained Transformers have emergent structures that find these binary representations and utilize them to make systematic predictions from the context. Notably, the per-digit period plays and important role in the prediction accuracy of that

*Figure 3.* Bit-wise periods in an example LCG sequence generated with $m = 2048, a = 293, c = 1033$, which follows the Hull-Dobell theorem. In binary representation $w$-th lowest bit has a period of $2^w$, for $w \in \{1, \ldots 11\}$. Writing a new sequence by skipping every 2nd step ($r = 2$) reduces the periods of all the bits by a factor of 2, rendering the lowest bit constant. $r = 2^k$ reduces bit-wise periods by a factor of $2^k$, with last $k$ digits constant.

digit. To understand this, consider the $r$-step iteration of Equation (1):

$$x_{t+r} = a^r x_t + \sum_{i=1}^{r} a^{i-1} c \quad \bmod m . \quad (3)$$

In this new sequence wherein we skip $r$ steps, the period of each digit $\alpha_{w-1}$ reduces from $2^w$ to $2^w / \gcd(r, 2^w)$. Consequently, the higher digits become relatively simpler to predict due to reduced periods while some lower digits become trivial to predict due to being constant along this new sequence. We demonstrate this for $m = 2048$ in Figure 3 (top panel). The digit-wise periods in the new sequence with $r = 2$ are reduced by a factor of 2, while the last digit is simply constant. Higher values of $r = 2^k$ will lead to even further simplifications of the sequence. Transformers can simplify the task of predicting LCG sequences by utilizing $r$-step iterations from the in-context examples – with longer contexts leading to larger values of $r$. Consequently, the per-digit and overall accuracies improve substantially with context (see Figure 4).

While moduli of the form $m = 2^{(\cdot)}$ lead Transformers to find binary representations, similar simplifications in composite moduli require more general representations of the *Residual Number System (RNS)* (Garner, 1959). RNS represents numbers by their values modulo pairwise co-prime factorizations of $m$. Specifically, consider sequences with a composite modulus $m$, which has a prime factorization $m = p_1^{w_1} p_2^{w_2} \cdots p_q^{w_q}$. In this case, we can uniquely represent each number ($x \bmod m$) as the tuple of residuals ($x \bmod p_1^{w_1}, x \bmod p_2^{w_2}, \ldots, x \bmod p_q^{w_q}$). Analogous to Equation (2), we can further decompose each residual,

$$x \bmod p_j^{w_j} = \alpha_{j,0}\, p_j^0 + \alpha_{j,1}\, p_j^1 + \cdots + \alpha_{j,w_j-1}\, p_j^{w_j-1} \tag{4}$$

where $\alpha_{j,w} \in \{0, 1, \ldots, p_j - 1\}$ are base-$p_j$ digits. We refer to $\{\alpha_{j,w}\}$ as the "RNS representation" in the remainder of the text. When $\mathcal{T}_m = m$, each digit $\alpha_{j,w}$ has a period of $p_j^w$ (derivation in Appendix F.2). The $r$ step iteration Equation (3) reduces the period of each digit $\alpha_{j,w}$ from $p_j^w$ to $p_j^w / \gcd(r, p_j^w)$. This results in simplification of the prediction task whenever $r = p_1^{k_1} p_2^{k_2} \cdots p_q^{k_q}$. We will see that identifying the RNS representations is a key simplification that the Transformer discovers in learning LCG sequences.

## 4.2. Interpretability: Fixed Modulus

Qualitative Algorithm (fixed modulus):

i. Find RNS representations of inputs from the learned prime factorization of $m$

ii. Look back $r = p_j^k$ steps in the context and copy the lowest $k$ digits, for different prime factors ($p_j$) of $m$

iii. Using these $r$-step iterations, predict the higher digits of the simplified sequence

We now discuss the algorithm implemented by Transformers trained on LCG with a fixed modulus. We will show that, despite being provided integers as inputs, the model develops emergent structures that create and leverage the RNS representation of the inputs. We will focus on the setup with a 1-layer, single-head Transformer trained on LCG with $m = 2048$. Similar results for composite moduli (e.g. $m = 7776$) and different model sizes are presented in Appendix G. We emphasize that this algorithm works for arbitrary $a, c, x_0$ for a given $m$.

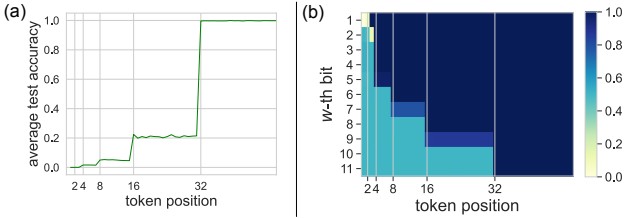

Figure 4. **FM:** Test accuracy for $m = 2048, \mathrm{depth} = 1, n_{\mathrm{heads}} = 1, d_{\mathrm{model}} = 768$, averaged over $a$, $c$, and seeds. (a) Test accuracy w.r.t. token positions. Ladder-like structure appears, with jumps occurring at $2^k$-th positions. (b) We represent numbers as an eleven-digit binary number ($2048 = 2^{11}$) and compute the per-digit test accuracy of model predictions.

We begin by analyzing the average accuracy of a trained

Transformer model as a function of token position along the context, shown in Figure 4 (a). (Recall that in this section a token corresponds to an integer of the LCG sequence). The accuracy exhibits a ladder-like structure, with each jump occurring exactly at the $2^k$-th token position. These successive transitions can be explained using binary representations and $r$-step recurrence (Equation (3)). Specifically, to predict the token at position $t \geq 2^k$, the model can look back in the context at position $t - 2^k$ and implement $r = 2^k$-step iteration. This allows the model to (i) copy the lowest $k$ bits since they remain unchanged; and (ii) simplify the higher bits, since their periods get reduced by a factor $2^k$. We note that the accuracy trend remains unchanged across different choices of $a$ and $x_0$ (see Figure 21).

Next, in Figure 4(b), we compute the per-digit accuracy by converting both model predictions and ground truth labels to their binary representations according to Equation (2). We observe that the model can predict more digits correctly with more context, with sharp transitions occurring at $2^k$-th token positions. This is a direct result of the sequences becoming increasingly more simplified as the model can look farther back in the context and utilize ($r = 2^k$)-step iterations. Since these simplifications occur in the form of digit-wise periods, Figure 4(b) serves as direct evidence that the model is internally developing and utilizing binary representations. The sequential learning of digits also explains the ladder-like structure of the accuracy in Figure 4(a). We find that the overall average accuracy (Figure 4(a)) multiplicatively depends on the per-digit accuracies (Figure 4(b)) (empirical proof in Figure 22(b)) [1]

$$\mathrm{acc}_{\mathrm{overall}} = (\mathrm{acc}_{\mathrm{digit}\,1})(\mathrm{acc}_{\mathrm{digit}\,2}) \cdots (\mathrm{acc}_{\mathrm{digit}\,11}). \tag{5}$$

Next, we investigate how various components of the model implement the algorithm outlined earlier this section.

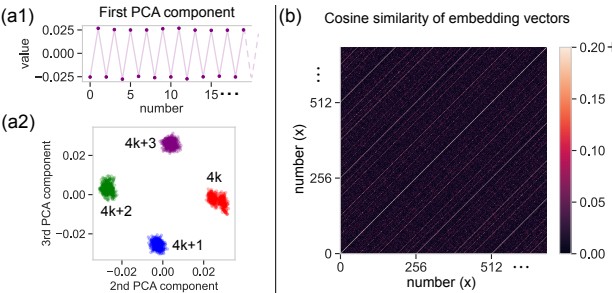

Figure 5. **FM:** ($a = 1589, c = 629$) Embedding layer. (a1) 1st principal component groups the numbers mod 2. (a2) 2nd and 3rd principal components group numbers mod 4. (b) Embedding vectors of different numbers exhibit high cosine similarity when they are spaced $2^k$ apart, with the similarity increasing with $k$.

[1]This holds if the per-digit accuracies are independent, which we confirm empirically.

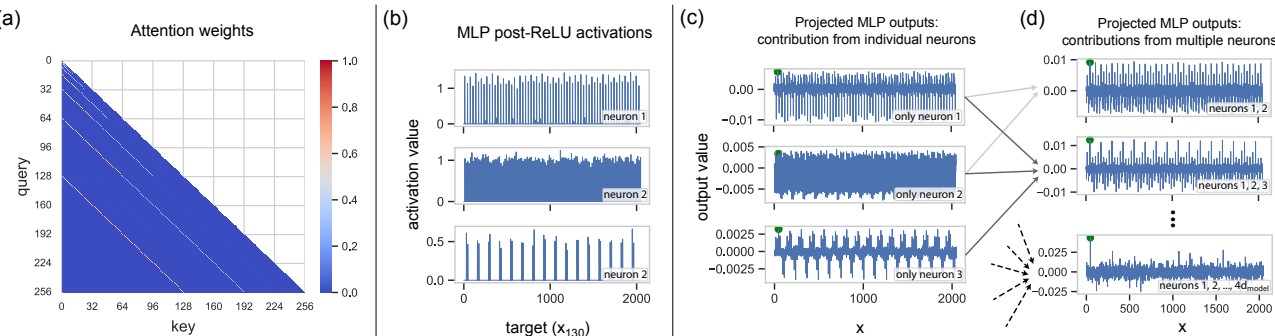

*Figure 6.* **FM:** ($m = 2^{11}, a = 1589, c = 629$) (a) Attention weights: each query attends most strongly to the tokens $2^k$ and $2^{k-1}$ distance backward, for the highest possible value of $k$, enabling copying of lowest $k$ bits. The other faint lines facilitate the prediction of higher bits. (b) Post-ReLU hidden layer MLP activations at token position $t = 129$ (extracted using sequences with different $x_0$) as a function of the target number $x_{130}$ which it is supposed to predict. Each neuron gets activated only while predicting a specific $x_{130}$, exhibiting a sparse, periodic pattern. (c) Output of the MLP block projected onto the (un)embedding matrix; after masking out all but a single given hidden-layer neuron. The green dot denotes the value at the target number. Each neuron resolves the correct prediction up to a periodic structure. (d) Output of the MLP block projected onto the (un)embedding matrix; after combining the signal from multiple neurons (i.e. gradually un-masking the neurons). The per-neuron periodic patterns constructively interfere at the correct output.

**Step i:** We begin by conducting Principal Components Analysis (PCA) of the embedding matrix, which shows how the model performs prime factorization to develop the binary representations (RNS for general $m$). Figure 5(a1) shows the projections of all numbers $x \in \{0, \dots 2047\}$ along the first principal component of the embedding. We observe that the model groups the numbers into modulo 2 clusters along the first principal component. Similarly, the 2nd and 3rd principal components group the numbers into modulo 4 clusters (Figure 5(a2)). In general, we find principal directions that group the numbers into modulo $2^{(\cdot)}$ (see Figure 24). By clustering the numbers according to their remainder modulo different prime-powers, these principal directions naturally encode the digit-wise representations of the inputs. In Figure 5(b), we check the cosine similarity between the embedding vectors of different numbers. We see that the more digits two numbers share in the binary representation, the higher the cosine similarity of their embedding. This is a consequence of these numbers having similar components along principal directions corresponding to those digits. For composite moduli, we find similar clustering according to different prime factors of $m$ (see Figures 25, 26).

**Step ii, iii:** In Figure 6(a) we examine the attention weights and find that to predict the number at position $t$, the model attends most strongly to the position $t - 2^k$ for the highest possible value of $k$ s.t. $t \geq 2^k$ (i.e. $k = \lfloor \log_2 t \rfloor$). This corresponds to the brightest line in Figure 6(a). Using the binary representation of the $(t - 2^k)$-th token, the model can copy the lowest $k$ bits and simplify the prediction of higher bits. Additionally, the second brightest line appears at the position $t - 2^{k-1}$, along with other faint lines at intermediate distances (multiples of $t - 2^{k'}$ for $k' < k - 1$). The information obtained from all these lines are utilized by

the model in predicting the higher bits.

To verify that the brightest line enables the copying of lower bits and that the second brightest line facilitates the prediction of higher bits, we performed the following two experiments. (i) For each query in the attention head, we mask out all keys except for the one at position $t - 2^k$. We then measure the performance of this ablated model, shown in Figure 20(a). We observe that the model retains the ability to copy the $k$ lowest bits, but loses the ability to predict the higher bits. (ii) Next, we repeat the above ablation experiment while masking out all keys except the ones at positions $t - 2^k$ and $t - 2^{k-1}$, shown in Figure 20(b). This results in a drastic improvement in the prediction of the higher bits compared to (i), confirming our assertions.

After the attention layer collects information about previous tokens, the MLP block [2] processes the information to make predictions. We find that each hidden neuron (post-ReLU) in the MLP exhibits a periodic response with a distinct period, as a function of the target prediction. Consider the MLP at token position $t$. The input to the transformer at this position is $x_t$ and the output $y_t$, should match the next number in the sequence $x_{t+1}$. In Figure 6(b) we show the activation value of selected neurons at token position $t = 129$ as a function of the target $x_{130}$, for $m$ different sequences obtained by changing the seed $x_0$ for a given LCG sequence. We observe that only a sparse set of neurons are activated for a given target $x_{130}$ and that there is a strong spiked periodic structure in the response of each neuron as a function of $x_{130}$, with neuron-specific frequencies.

The subsequent fully connected layer in the MLP block aggregates the contributions from all the activated neurons to

---

[2]MLP block: fully connected $\rightarrow$ ReLU $\rightarrow$ fully connected.

make the correct prediction (McCracken et al., 2025). To visualize the contributions from an individual neuron, we mask out the contribution from all other neurons and extract the MLP output for a fixed input sequence. We then project this output onto the (un)embedding matrix, which shows us the contribution of that single neuron in making the correct prediction.[3] In Figure 6(c) we show that each neuron resolves the target number $x_{130}$ up to a distinct periodic pattern. The periodic patterns from different neurons constructively interfere at the target. In Figure 6(d), we observe that gradually adding contributions from multiple neurons resolves the correct output with increasing accuracy.

### 4.3. Interpretability: Generalization to Unseen Modulus

---

Qualitative Algorithm (unseen modulus):

  i. Encode information about various possible prime factorizations

  ii. Estimate the modulus via the largest number in context

  iii. Combine steps i and ii to construct correct RNS representations, then implement steps ii and iii from the fixed modulus algorithm

---

Unlike the **FM** case, the training set for **UM** is generated using many different moduli $m$, unseen at test time. Since each modulus has its own RNS representation incompatible with other moduli, the model must implement a different, more general algorithm to solve **UM** tasks.

We analyze how the embedding layer and attention heads of a 4-layer Transformer model help implement the above algorithm to solve the **UM** task. The model is trained on a dataset generated with $n_m = n_a = n_c = 128$, where $m_{\max} = 2457$. To avoid leakage between training and test sets, we specifically exclude the moduli $m_{\text{test}} \in \{1800 = 2^3 \cdot 3^2 \cdot 5^2, 2048 = 2^{11}, 2352 = 2^4 \cdot 3 \cdot 7^2\}$ from the training set. Note that the choice $m_{\max} = \lfloor 1.2 \cdot 2048 \rfloor$ is made to maintain consistency with the setting in Figure 1(a).

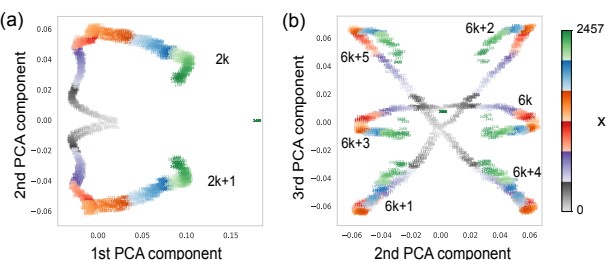

**Figure 7. UM**: PCA analysis of the embedding layer.

---
[3]We read-out MLP outputs instead of network outputs to avoid distortion from the skip connection.

**Step i:** We first analyze the embedding layer. In Figure 7(a), PCA shows a semi-circular structure along the first principal component, with the second principal component separating even and odd numbers. This semi-circle resembles the circular patterns seen in modular arithmetic tasks (Power et al., 2022; Zhong et al., 2023), but since our model is trained on multiple moduli, it cannot form a closed circle by identifying a unique 0 value. Figure 7(b) further shows that the 2nd and 3rd principal components group numbers by their remainders modulo 2 and 3, which likely reflects their prevalence as prime factors in the training set.

Note with varying moduli, the model need not form a binary encoding as before. Instead, we find attention heads in first layer group embedded numbers by remainders modulo different primes, each head specializing in a particular factor. This specialization allows the model to construct RNS representations with various prime bases.

To investigate head specialization, we analyze affermentioned first-layer attention heads in Figure 8, presenting PCA results for selected heads in panels (a1, b1, c1). For each, we input sequences using the corresponding $m_{\text{test}}$ and randomly selected $(a, c)$ pairs (per the Hull-Dobell Theorem), then isolate the output $\boldsymbol{H}^{(h)}$ for each head by zeroing out all other heads. We perform PCA at token position $t = 0$ ($\text{PCA}(\boldsymbol{H}^{(h)}[:, 0, :])$), then projecting each number's feature vector $\boldsymbol{H}^{(h)}[x, t, :]$ onto the top two principal components and labeling each point with its corresponding $x$.

The analysis shows that each head groups numbers by their remainder modulo different prime factors, enabling the model to select suitable representations in later layers. The prominence of small primes in the top principal components likely reflects their frequency in the training data. Furthermore, we believe the performance gains observed in Figure 1 with more heads can be partly attributed to the model's improved capacity to capture additional prime factors for constructing RNS representations. More examples of specialization across various $(a, c, m_{\text{test}}, x_0, t)$ are provided in Figure 27 (Appendix H.1).

To further demonstrate that these heads directly influence the model's performance, we measured per-digit accuracy before and after pruning each specialized head, as shown in panels (a2, b2, c2) and (a3, b3, c3). For pruning, we replace the head's output with its mean value $\text{mean}(\boldsymbol{H}^{(h)}[:, :, :]) \in \mathbb{R}$ across all positions (using 10% of randomly selected training sequences), which preserves signal scale and avoids catastrophic model degradation. The results show that pruning a head responsible for a particular prime factor significantly impairs performance on corresponding digits, while other digits are less affected. For instance, in panel (a3), the model's ability to compute $7^1$ and $7^2$ digits is lost, while base-2 and base-3 digits remain above chance. Similarly, in (b3), removing the modulo-2 head

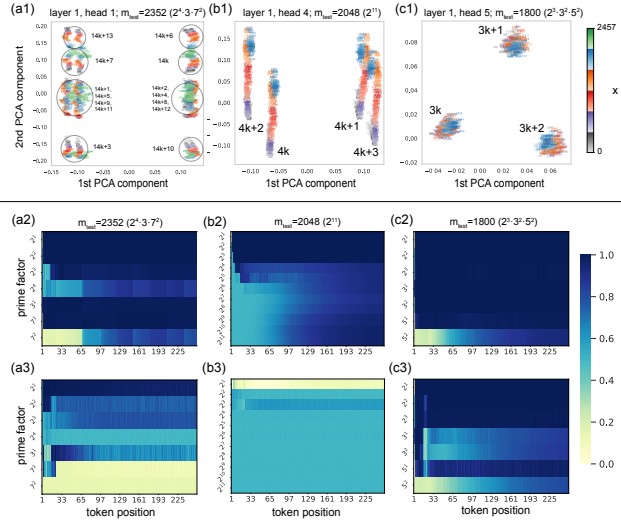

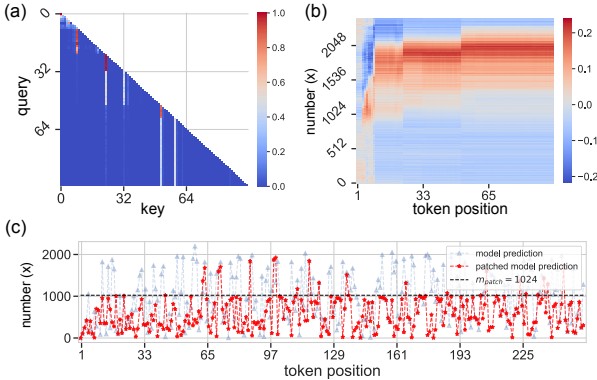

*Figure 9.* **UM**: Attention head (layer 1, head 6) specialized in estimating $m_{\text{test}}$. (a) Queries attend to largest keys; (b) the head produces features with the highest cosine similarity to $m_{\text{test}}$; (c) patching this head with features from sequences with $m_{\text{patch}} = 1024$ (same $a$, $c$) steers the model to predict numbers $n < m_{\text{patch}}$.

*Figure 8.* **UM**: PCA analysis of attention heads specialized for different prime factors and their impact on per-digit accuracy. (a1, b1, c1) PCA of outputs from specific heads; the first two principal components group numbers by remainder modulo 14, 4, and 3, respectively. (a2, b2, c2) Test accuracy for individual digits with representations from Equation (4) for each $m_{\text{test}}$. (a3, b3, c3) Per-digit test accuracy after pruning these heads, showing substantial performance degradation on the affected digits.

erases all corresponding digit accuracy, and in (c3), pruning the modulo-3 head greatly reduces base-3 digit performance, but leaves base-2 and base-5 largely intact.

Finally, we further validate the link between head specialization and digit-wise accuracy by pruning irrelevant heads. For $m_{\text{test}} = 2048$, pruning heads responsible for modulo 7 or 3 (as in panels (a1) or (c1)) yields the results in Figure 28. Interestingly, removing the modulo-3 head sometimes improves performance for specific token positions, while others show minimal degradation. These findings reinforce the connection between head specialization and digit-wise computation; further details can be found in Appendix H.2.

**Step ii:** The performance variations observed above suggest that the model is internally uncertain about which RNS representation to use, likely due to difficulty in determining $m_{\text{test}}$. As we discuss below, the model appears to estimate $m_{\text{test}}$ greedily via in-context learning.

In Figure 9(a), using a sequence with $x_0 = 1$, $a = 5$, $c = 31$, and $m_{\text{test}} = 2048$, we observe a first-layer attention head that attends primarily to the largest numbers, as seen by the vertical lines in the attention weights. To analyze this further, in panel (b), we extract the output $\boldsymbol{H}^{(h)} \in \mathbb{R}^{1 \times L \times d_{\text{model}}}$ from this head for the same sequence and compute its cosine similarity with token embeddings for all $x < m_{\text{test}}$, producing a $\mathbb{R}^{L \times m_{\text{test}}}$ matrix for the heatmap. This reveals that the head's output consistently has the highest similarity with the largest numbers that has been seen in context,

indicating a greedy estimation of $m_{\text{test}}$.

To further verify that this head estimates $m_{\text{test}}$ for later layers, we conduct a patching experiment (Zhang & Nanda, 2024), as shown in Figure 9(c). We generate a new sequence with the same $a$, $c$, and $x_0$, but with $m_{\text{patch}} = 1024$, and extract $\boldsymbol{H}^{(h)}\text{patch} \in \mathbb{R}^{1 \times L \times d\text{model}}$ from this head. We then overwrite the output $\boldsymbol{H}^{(h)}$ in a forward pass for the original sequence ($m_{\text{test}} = 2048$) with $\boldsymbol{H}^{(h)}\text{patch}$. The model now frequently predicts numbers smaller than $m\text{patch}$. As shown in Figure 30 (Appendix H.3), patching other heads disrupts predictions but never induces a similar qualitative shift, confirming this head's unique role in estimating $m_{\text{test}}$.

One might expect that even a small error in estimating $m_{\text{test}}$ would invalidate predictions, but this is not necessarily the case. As shown in Section 4.2, if the correct representation is chosen, the lower bits maintain a strong periodic signal. Since the model can prepare multiple RNS representations (as demonstrated in **step i**), a sufficiently close estimate of $m_{\text{test}}$ allows these lower bits to guide the model toward the correct representation. Further discussion is provided in Appendix H.4 and **step iii**.

**Step iii:** Once the necessary features are prepared, subsequent layers implement the rest of the algorithm. As in the **FM** case, we observe a ladder pattern in digit-wise accuracy for lower digits at early token positions. This pattern, which demonstrates the copying bias, is visible in Figure 8(b2).

More careful inspection of Figure 8(b2) shows the model copies the lowest 5 digits. Interestingly, given the estimated $m_{\text{test}} = 2033$ from **step ii**, the model effectively reduces it to $2033/2^5 \approx 63.53$. Futhermore, since the model operates on integers, it could round it to 64, yielding $64 \cdot 2^5 = 2048 = m_{\text{test}}$. Thus, the model in principle, can predict higher digits accurately without the exact value of $m_{\text{test}}$.

While some algorithmic transition from lower to higher digits clearly occurs, the exact mechanism remains unclear. The sharp transition in per-digit performance at the $2^5$ digit in Figure 8(b2) supports this argument. Nevertheless, we believe the model applies an algorithm similar to **FM** for higher bits, which we will elaborate on in Appendix H.5.

## 5. Scaling Up the Modulus

In this section, we investigate training upon scaling up the LCG modulus, with the following modifications:

**Base-b tokenization:** To avoid massive dictionary sizes for large $m$, we implement a base-$b$ tokenization scheme. Each integer is decomposed into a sequence of base-$b$ digits, beginning with the least significant digit; resulting in a vocabulary size of $b$ for any $m$ (for details see Appendix I.1). Based on the discussion in Section 4.1, it is beneficial to choose $b$ such that $\gcd(b, m) = b$.

**Abacus Embeddings:** We encode positional information using a variant of the Abacus embedding (McLeish et al., 2024), as a sum of two learnable vectors. One vector encodes the position of the integer within the sequence, while the other encodes the position of each digit within the integer (for details, see Appendix I.2).

**Fixed Modulus:** For each modulus $m = 2^k$, where k is an integer in the range $16 \leq k \leq 32$, we train a 2-layer model with $d_{\text{model}} = 1024$ and a vocabulary size of 256. We select training sequences via the Hull-Dobell theorem, setting $n_a = n_c = 1024$ (See Appendix I.3 for training details). For the test dataset, we choose 512 values of $a$ and 64 values of $c$ that differ from those in the training set.

The quality of an LCG largely depends on its multiplier, traditionally evaluated via the spectral test (Knuth, 1997). In Figure 10(a), we test our model on both spectrally optimal Steele multipliers (Steele & Vigna, 2021) and arbitrary multipliers for $m = 2^{32}$. While achieving 100% test accuracy with equal in-context sequence lengths, the model performs consistently worse on Steele-generated sequences compared to those from arbitrary multipliers.

In Figure 10(b), a log-log plot reveals that the number of in-context sequence elements needed for 100% test accuracy scales sublinearly with modulus $m$ as $m^{\gamma}$, where $\gamma \approx 1/4$.

**Unseen modulus:** For the **UM** case, we train a 6-layer Transformer on a dataset with $n_m = 32{,}768$, $n_a = 128$, $n_c = 1$, with $1024 < m_{\text{train}} < 65{,}536$. Sequences are length 512, each integer tokenized as two bytes (context length 1023). As before, test data uses unseen $m_{\text{test}}$ and $(a, c)$, focusing on $m_{\text{test}} = 2^k, , 3^k$. Figure 11 shows that the number of in-context sequence elements needed to reach 60% test accuracy scales as $m_{\text{test}}^{\gamma}$ ($0.24 \leq \gamma \leq 0.33$). The averaged test accuracy of each number in the sequence is shown in

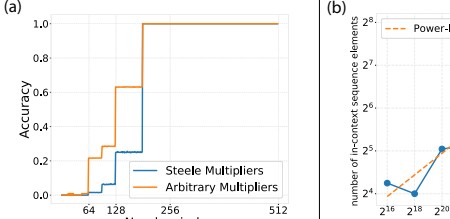
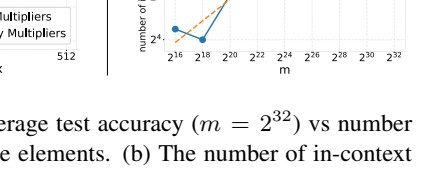

*Figure 10.* **FM**: (a) Average test accuracy ($m = 2^{32}$) vs number of in-context sequence elements. (b) The number of in-context sequence elements required to achieve 100% test accuracy (minimum of 5 runs). (See Appendix I.3 for details)

Appendix I.4. Test performance is influenced by the tokenization base, since the tokenization base highlights the periodic structure of LCGs making it more apparent and easier for the model to leverage during training and prediction. To confirm this, we train a model with tokenization base 243. In Figure 11, $m_{\text{test}} = 2^k$ ($3^k$) sequences scale better when the tokenization base is $256 = 2^8$ ($243 = 3^5$).

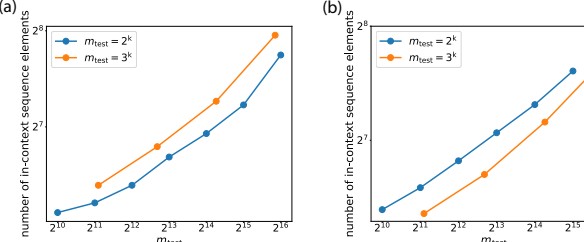

*Figure 11.* **UM**: The number of in-context sequence elements needed for 60% test accuracy grows sublinearly with modulus $m$, depending on compatibility between $m$ and tokenization. (a) Base-$2^8$ tokenization; (b) base-$3^5$ tokenization.

## 6. Conclusion

We have investigated Transformer training on LCG sequences, focusing on fixed modulus training as well as generalization to unseen moduli. In both cases, we have uncovered the algorithm used by the model to solve these tasks and highlighted the model components that implement the steps of the algorithm. We have found that the model finds and utilizes prime factorizations of $m$ and RNS representations of numbers to simplify the sequences and make predictions. We have provided the modified training recipe for scaling up the modulus in both **FM** and **UM** settings, and shown their scaling behaviors.

**Limitations and future work:** The results of this paper were limited to scales $m \leq 2^{32}$. It would be interesting to test our results on much larger moduli as well. We leave the exploration of PRNGs that are built upon LCGs, such as PCGs and truncated LCGs for future works. It would also be interesting to make the training even more unbiased, by training on general classes of arithmetic sequences.

## Acknowledgements

M.B. thanks Carl Miller for discussions on PRNGs. This work is supported by NSF DMR-2345644 (D.S.K., T.T., and M.B.), and by an NSF CAREER award DMR-2045181 (T.H. and D.D.). The authors acknowledge the University of Maryland supercomputing resources (http://hpcc.umd.edu) made available for conducting the research reported in this paper.

## Impact Statement

Our work advances the understanding of how neural networks learn deterministic sequences, specifically LCGs. While this capability cannot compromise mainstream cryptographic systems, which use far more sophisticated techniques, our insights may contribute to the development of more robust cryptographic algorithms and a better understanding of neural networks' computational capabilities.

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

# A. Experimental Details

This section provides further details about model architecture, dataset construction, and optimization.

## A.1. Dataset Construction

**Fixed Modulus (`FM`):** Given a modulus $m$, we apply the Hull-Dobell Theorem to determine the possible values of $(a, c)$ that maximize the period. We then randomly select $64$ values of $a$ and $c$ to generate the test dataset. To generate the training dataset, we *exlude* these test choices of $(a, c)$ and uniformly sample $N = 100,000$ LCG sequences of length $L$ (context length) with $n_a$ values of multipliers and $n_c$ values of increments. For each set of parameters $(a, c)$, we sample an LCG sequence with a randomly selected initial seed $x_0$. Note that the training dataset includes sequences with varying periods, while the test data only contains sequences that maximize the period.

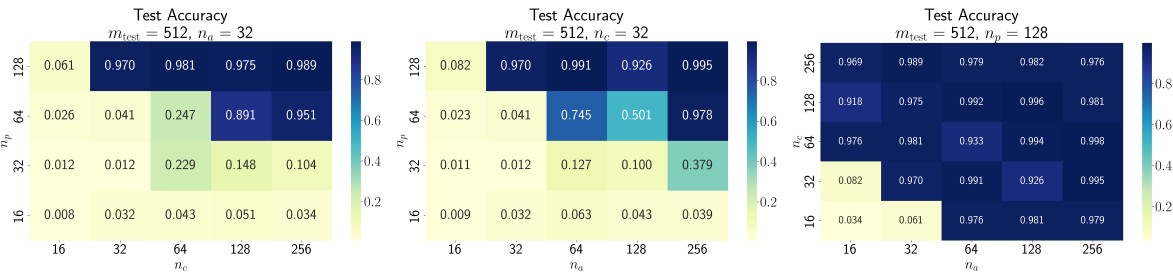

*Figure 12.* The impact of training dataset parameters $(n_m, n_a, n_c)$ on unseen modulus task performance.

**Generalization to Unseen Modulus (`UM`):** For the test dataset, we first select a set of test moduli $M_{\text{test}} = \{m_{\text{test}}\}$ that would be reserved exclusively for evaluation. For each test modulus $m_{\text{test}} \in M_{\text{test}}$, we apply the Hull-Dobell Theorem to determine the values of $(a, c)$ that maximize the period. We then randomly select $64$ values of $a$ and $c$ to generate the test dataset. These $64$ $(a, c)$ values are not considered while generating the training dataset.

For the training dataset generation, we sample the $n_m$ modulus values from the range $[L, \lfloor 1.2 \max(M_{\text{test}}) \rfloor]$ while excluding all the values in $M_{\text{test}}$. For each modulus value $m$, we uniformly select $n_a$ multipliers ($0 < a < m$) and $n_c$ increments ($0 \leq c < m$), excluding the ones reserved for testing. For each parameter $(a, c, m)$, we generate a sequence of length $L$ using a randomly selected initial seed $x_0$. This results in a total of $N = n_m \times n_a \times n_c$ training sequences. We report that $N = 400,000$ served sufficient from the modulus values considered in this work.

Next, we examine the effect of training dataset composition $(n_m, n_a, n_c)$ on the performance. Figure 12 shows the test accuracy primarily depends on $n_p$ and marginally on $n_a$ and $n_c$. Furthermore, we found that $n_m \gtrsim m_{\text{test}}/4$ yields good generalization performance (result not shown here). Based on this relationship and our target total number of training examples $N$, we sample $n_a = n_c = \sqrt{\frac{N}{m_{\text{test}}/4}}$ values of multipliers and increments. Unless explicitly specified, we use these parameter settings as the default configuration for all experiments.

## A.2. Model Architecture Details

We consider GPT-style Transformers (Radford et al., 2019) with learnable positional embeddings and weight tying (Press & Wolf, 2017). The model architecture is characterized by the number of blocks (depth), embedding dimension ($d_{\text{model}}$), and number of attention heads ($n_{\text{heads}}$). For most experiments, we use GELU activations, except in Section 4, where we use ReLU activations for better interpretability.

## A.3. Training Details

We train the models with Cross-entropy loss using AdamW optimizer (Loshchilov & Hutter, 2019) with momentum hyperparameters $\beta_1 = 0.9$ and $\beta_2 = 0.99$. We implement a linear learning rate warmup over the first $2048$ steps with an initial learning rate of zero and the target learning rate $\eta$. By default, all experiments employ a batch size of $256$. Weight decay is only applied to non-bias parameters.

In the unseen modulus case, we observed that both the optimal target learning rate $\eta$ and weight decay strength $\lambda$

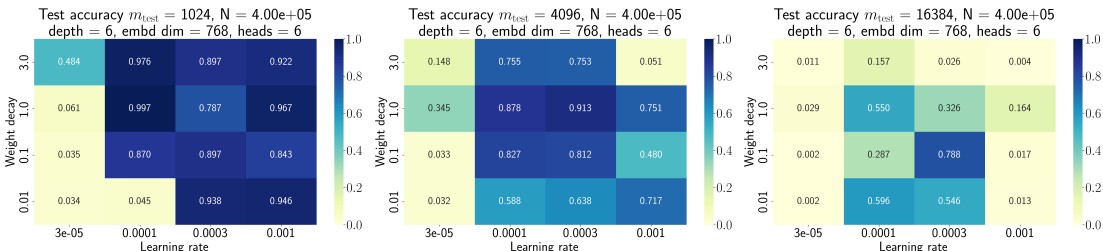

*Figure 13.* Heatmap of test accuracy of a 6 layer Transformer with learning rate and weight decay as the axes. As the modulus is increased from 1024 to 16, 384 the range of hyperparameters resulting in reasonable accuracy becomes narrow.

are highly sensitive to minute changes in training dataset properties (modulus $m_{\text{test}}$, number of LCG parameters $n_m$, $n_a$, $n_c$ and total number of examples $N$) and architectural changes (depth and embedding dimension). To determine the optimal hyperpamraters, we scan the learning rates $\eta \in \{3e\text{-}05, 1e\text{-}04, 3e\text{-}04, 1e\text{-}03\}$ and weight decay strengths $\lambda \in \{0.01, 0.1, 1.0, 3.0\}$.

### A.4. Training Cost

In Figure 10, the training of the $m = 2^{32}$ model was conducted using four NVIDIA A100 GPUs, requiring a total of 21.82 hours. The $m = 2^{16}$ model completed training in 4.83 hours under the same hardware setup. Despite having the same model size, the increased context length in the $m = 2^{32}$ model led to a significantly higher computational cost.

In Figure 11, both models were trained on a single NVIDIA H100 GPU for 22 hours.

### A.5. Hyperparameter Space Shrinking with increasing modulus

We also report a surprising phenomenon in the unseen modulus case, which we refer to. as the 'hyperparameter space shrinking.' As we increase the test modulus $m_{\text{test}}$ while keeping the model architecture fixed, we observe that the optimal learning rate ($\eta$) and weight decay strength ($\lambda$) shift significantly. Moreover, the range of these hyperparameter values that yield reasonable performance becomes increasingly narrow. Figure 13 shows this result for a 6 layer Transformer.

## B. Fixed Modules Training Results

For the **FM** case, we find that a single attention head in one layer is sufficient to solve the task when the modulus is not a prime number. In Figure 14, we present the model's training loss and performance across training steps for $m = 2048$. Notably, panels (c, f) reveal a significant disparity between training and test loss, indicating a grokking transition during the training process. Similarly, we plot in Figure 15 for similar curves for $m = 7776$, where all curves are qualitatively the same.

## C. Prime Moduli

In the **FM** setting, when $m$ is a prime number, the task becomes much harder. Since there are no digit-wise periodic patterns, the model cannot perform the algorithm described in Section 4.2. In Figure 16, we trained two identical models to learn $m = 2039$ and $m = 2048$, and we observe that the task with $m = 2039$ cannot be learned within the same number of training steps. Note that to rule out potential constraints from model capability, we used depth 2 models (as oppposed to depth 1 in the main text) in both cases.

In the **UM** setting, the model exhibits similar test performance on sequences with prime $m$ and sequences with $m$ as a power of two, as shown in Figure 17. We hypothesize that training on a diverse set of moduli helps the model rely less on the digit patterns and instead focus on more generalizable structure.

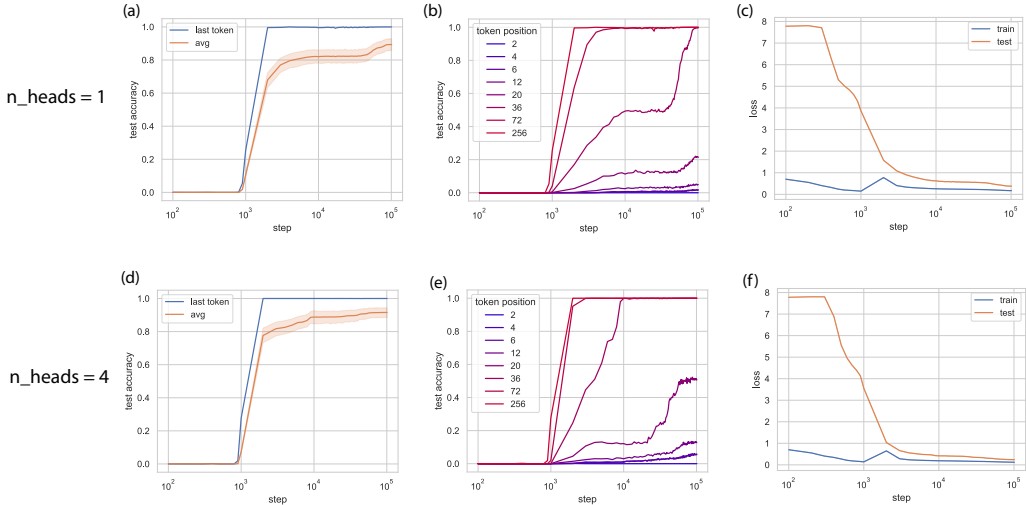

*Figure 14.* Test accuracy and train/test loss for $m = 2048 = 2^{11}$, depth=1. (a,b,c) $n_{\mathrm{heads}} = 1$ (d,e,f) $n_{\mathrm{heads}} = 4$.

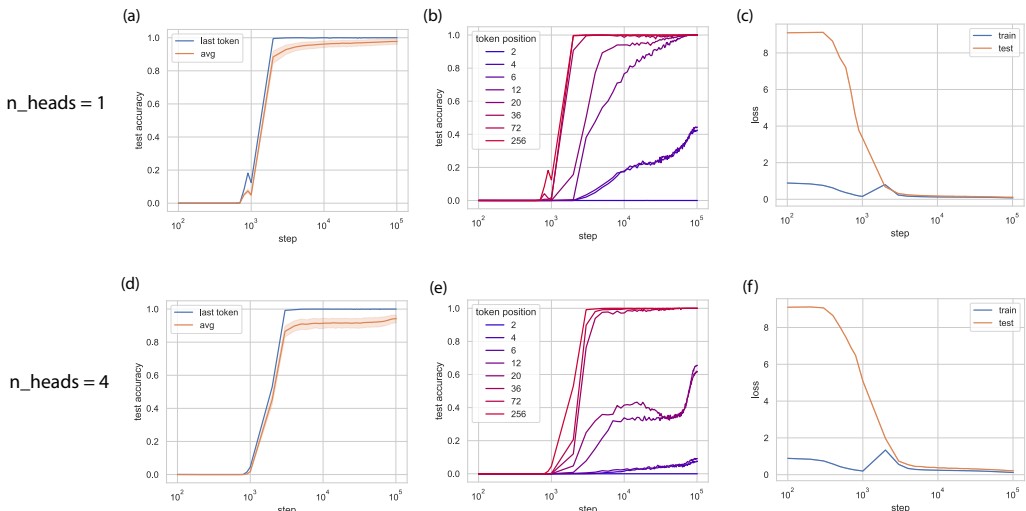

*Figure 15.* Test accuracy and train/test loss for $m = 7776 = 2^5 \cdot 3^5$, depth=1. (a,b,c) $n_{\mathrm{heads}} = 1$ (d,e,f) $n_{\mathrm{heads}} = 4$.

## D. Critical Depth for the Unseen Modulus Task

This section analyzes the depth and embedding dimension requirements for successfully training a Transformer on the unseen modulus task. The experimental details are the same as described in Appendix A.

We varied depths $\in \{2, 3, 4, 6, 8\}$ and embedding dimensions in $d_{\mathrm{model}} \in \{512, 768, 1024, 1280\}$, with the head dimension fixed to $d_{\mathrm{head}} = 128$. For each depth and width, we scanned learning rates $\eta \in \{3e\text{-}05, 1e\text{-}04, 3e\text{-}04, 1e\text{-}03\}$ and weight decay strengths $\lambda \in \{0.01, 0.1, 1.0, 3.0\}$ to identify the optimal hyperparameters. We report that the optimal learning rate and weight decay strength heavily vary with depth, embedding dimension, and training dataset. For $m_{\mathrm{eval}} = \{1024, 4096\}$, the models were trained for $T = 100,000$ steps, while for $m_{\mathrm{eval}} = 16,384$, the models required a longer training for $T = 200,000$ steps.

Figure 18 shows the test accuracy heatmaps with depth and embedding dimensions as the two axes. These results demonstrate that a minimum depth of 3 is required to learn the LCG sequence prediction task, with a marginal dependence on embedding dimension. This suggests the unseen modulus task requires a minimal computational depth of three to capture the underlying structure of LCGs.

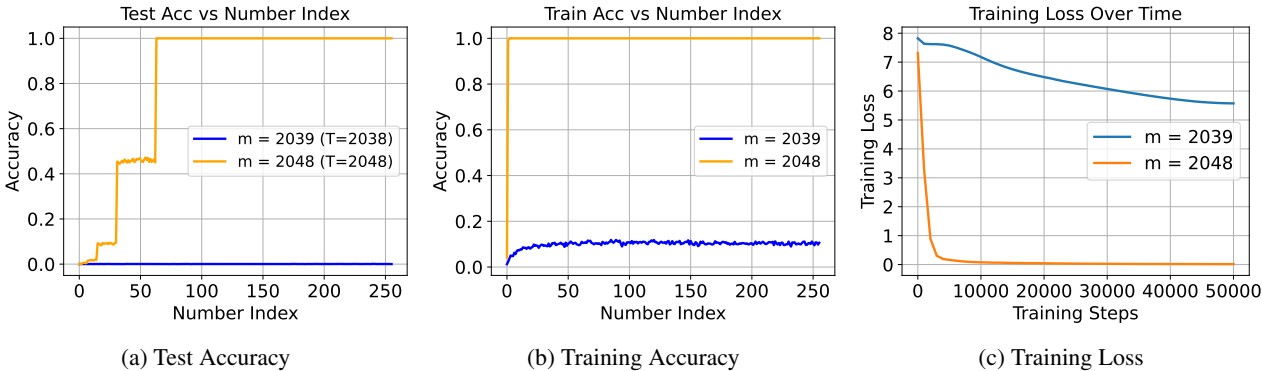

*Figure 16.* **FM:** Comparison between $m = 2039$ (prime) and $m = 2048$ (power-of-two), with both models trained for 50,000 steps. Each model has depth 2 and $d_{\text{model}} = 1024$. The test set consists of sequences with the same periods, while the training set includes arbitrary multipliers not present in the test set.

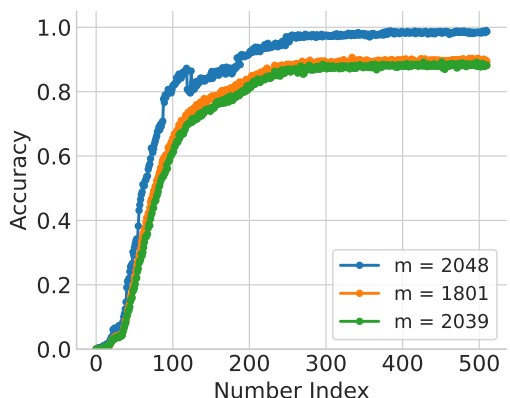

*Figure 17.* **UM:** Test accuracy comparison between $m = 2039$ (prime), $m = 1801$ (prime) and $m = 2048$ (power-of-two). The model was trained for 100,000 steps on a dataset consisting of 262,144 sequences with 512 distinct training moduli not present in the test set.

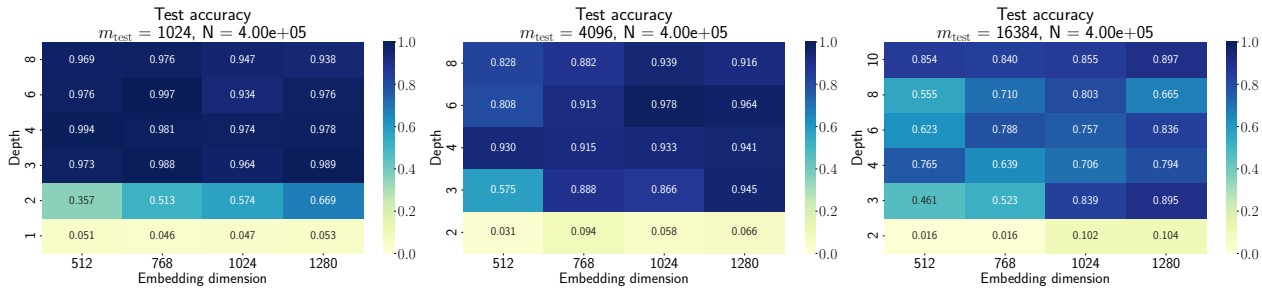

*Figure 18.* Test accuracy heatmaps of with depth and embedding dimensions as the two axes.

## E. Training Time Interpretability

In this section, we examine the order in which training sequences with different periods are learned during training. For this experiment, we consider a six-layer Transformer with $4$ heads and an embedding dimension of 768. For this experiment, we generate sequences of length 512 using 128 unseen moduli and 128 values of $a$ and $c$ each. The model is trained with Adam hyperparameters: $\eta = 3 \times 10^{-4}$ $\beta_1 = 0.9, \beta_2 = 0.99$ and weight decay strength $\lambda = 1.0$.

Figure 19(left) compares the training accuracy of sequences with different periods relative to the context length 512. We

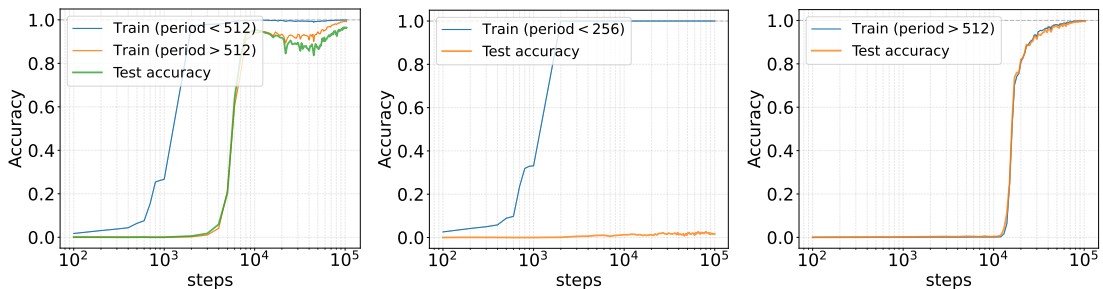

*Figure 19.* (left) Comparison of the training accuracy of sequences with different periods relative to the context length 512, (center) Accuracy when the model is only trained on sequences with period $< 512$, (right) Accuracy when the model is trained on sequences with period $> 512$.

observe that sequences with period $< 512$ are memorized early in training, while the accuracy of long-period sequences coincides with the test accuracy. Next, we perform two more experiments by training the model on datasets consisting of (1) sequences with period $< 256$, and (2) sequences with period $> 512$. Figure 19(center, right) show the results of these experiments. The model fails to generalize when trained only on low-period sequences while training only on long-period sequences eliminates grokking.

## F. LCG Properties

### F.1. The period of the lower $k$-th bit when $m$ is power of 2

In this section, we show that for a sequence of period $\mathcal{T}_m = m = 2^K$, the $k$-th lowest digit has a period of $2^k$ along the sequence. Consider an LCG sequence:

$$x_{t+1} = (ax_t + c) \mod m, \tag{6}$$

where $m$ is a power of 2, $m$ and $c$ are coprime and $a - 1$ is divisible by 4. The period of sequence $x_t$ is m (Hull & Dobell, 1962). The lower $k$-th bits of $x_t$ is given by:

$$b_{t,k} = \frac{z_{t,k} - z_{t,k-1}}{2^{k-1}}, \tag{7}$$

where $z_{t,k} = x_t \mod 2^k$. Therefore, if $z_{t,k}$ has a period of $2^k$, then the lower $k$ bits also have a period of $2^k$. Below, we show that $z_{t,k}$ has a period of $2^k$.

For an integer $M_t$, we can can re-write $x_t$ as:

$$x_t = z_{t,k} + M_t 2^k. \tag{8}$$

Next, we substitute Equation (6) into the definition of $z_{t+1,k}$:

$$
\begin{aligned}
z_{t+1,k} &= x_{t+1} \mod 2^k, \\
&= [(ax_t + c) \mod m] \mod 2^k.. 
\end{aligned} \tag{9}
$$

As $m$ is divisble by $2^k$, this simplifies to:

$$
\begin{aligned}
z_{t+1,k} &= (az_{t,k} + c) \mod 2^k, \\
&= (az_{t,k} + aM_t 2^k + c) \mod 2^k, \\
&= (az_{t,k} + c) \mod 2^k.
\end{aligned} \tag{10}
$$

Therefore, $z_{t,k}$ follows its own LCG recurrence with the same $a$ and $c$ but with a reduced modulus $2^k$. Because $2^k$ and $c$ are coprime and $a - 1$ is divisible by 4, the period of $z_n$ is $2^k$. Thus, the period of the lower $k$ bits is $2^k$.

Since $z_{n,k}$ has period $2^k$ and $z_{n,k-1}$ has period $2^{k-1}$, the period of $b_{n,k}$ is $2^k$.

### F.2. Derivation of Equation (4)

We now derive Equation (4) of the main text:

$$x_t \bmod p_i^{w_q} = \alpha_{i,0,t}\, p_i^0 + \alpha_{i,1,t}\, p_i^1 + \cdots + \alpha_{i,w_i-1,t}\, p_i^{w_i-1},$$

where $\alpha_{i,w,t} \in \{0, 1, \ldots, p_i - 1\}$ are base-$p_i$ digits. When the period of $x_t$ is m, each digit $\alpha_{j,w,t}$ has a period of $p_i^w$.

Consider an LCG sequence:

$$x_{t+1} = (ax_t + c) \mod m, \tag{11}$$

where $m$ has a prime factorization $m = p_1^{w_1} p_2^{w_2} \cdots p_q^{w_q}$, $m$ and $c$ are coprime, $a - 1$ is divisible by all prime factors of $m$ and $a - 1$ is divisible by 4 if $m$ is divisible by 4 (Hull & Dobell, 1962).

Consider the residulas $R_{i,t} = x_t \mod p_i^{w_i}$, where $i \in \{1 \ldots q\}$. We have:

$$x_t = R_{i,t} + M_{i,t} p_i^{w_i}, \tag{12}$$

where $M_{i,t}$ is an integer.

Substituting Equations (11) and (12) into the definition of $R_{i,t+1}$:

$$\begin{aligned}
R_{i,t+1} &= x_{t+1} \mod p_i^{w_i}, \\
&= [(ax_t + c) \mod m] \mod p_i^{w_i}, \\
&= (ax_t + c) \mod p_i^{w_i}, \\
&= (aR_{i,t} + aM_{i,t} p_i^{w_i} + c) \mod p_i^{w_i}, \\
&= (aR_{i,t} + c) \mod p_i^{w_i}. 
\end{aligned} \tag{13}$$

Next we consider $z_{i,k,t}$, which is the lower $k$ base-$p_i$ digits of $R_{i,t}$:

$$z_{i,k,t} = R_{i,t} \mod p_i^k. \tag{14}$$

Similarly, the recurrence of $z_{i,k,t}$ can be simplied as:

$$\begin{aligned}
z_{i,k,t+1} &= R_{i,t+1} \mod p_i^k, \\
&= [(aR_{i,t} + c) \mod p_i^{w_i}] \mod p_i^k, \\
&= (aR_{i,t} + c) \mod p_i^k, \\
&= (az_{i,k,t} + c) \mod p_i^k. 
\end{aligned} \tag{15}$$

$z_{i,k,t}$ follows an LCG recurrence with the same $a$ and $c$ and a reduced modulus $p_i^k$. Because $p_i^k$ and $c$ are coprime and $a - 1$ is divisible by $p_i$, the period of $z_{i,k,t}$ is $p_i^k$.

$\alpha_{i,k,t}$, which is the lower $k$-th base-$p_i$ digits of $y_{i,t}$, can be written as:

$$\alpha_{i,k,t} = \frac{z_{i,k,t} - z_{i,k-1,t}}{p_i^{k-1}}. \tag{16}$$

Since $z_{i,k,t}$ has period $p_i^k$, the period of $\alpha_{i,k,t}$ is $p_i^k$.

## G. Fixed Modulus Interpretability

In this subsection, we show additional results on the model's behavior on the **FM** task.

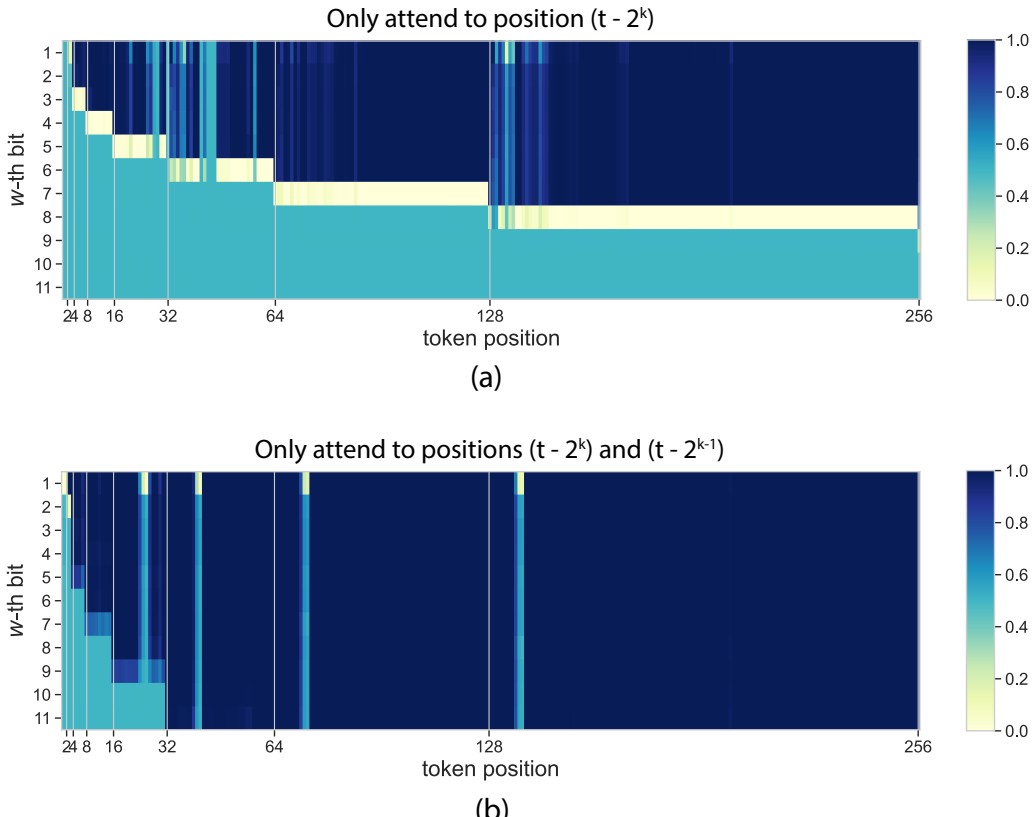

*Figure 20.* ($m = 2048, l = 1, n_{\text{head}} = 1, d_{\text{model}} = 768$) Per-digit test accuracies in binary representation, with attention masking (at inference time). $m = 2048 = 2^{11}$, depth=1, $n_{\text{heads}} = 1, d_{\text{model}} = 768$, averaged over $a, c$ and initial seeds. (a) For each query, only the attention to key at position $t - 2^k$ (where $k = \lfloor \log_2 t \rfloor$) is kept. All other attention weights are masked out. In this case, the model can successfully copy the last $k$ bits, but cannot predict higher bits. (b) For each query, only the attention to key at positions $t - 2^k$ and $t - 2^k$ are kept. All other attention weights are masked out. In this case, the model can predict higher bits with fairly high accuracy – this is remarkable given that majority of the attention weights are masked out.

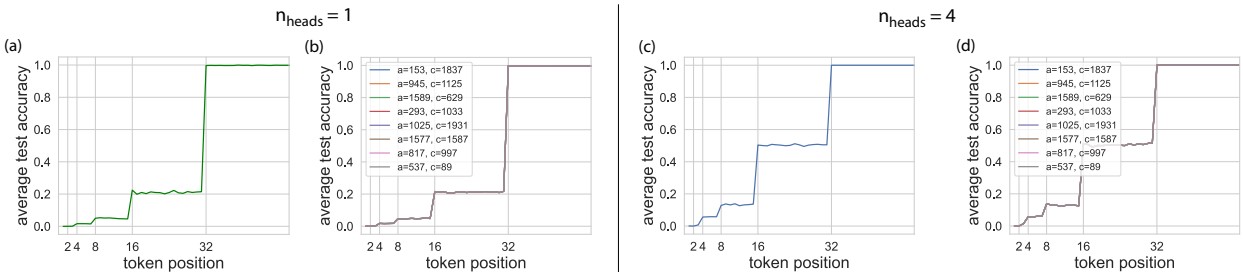

*Figure 21.* Test accuracy vs token positions for $m = 2048 = 2^{11}$, depth=1, various values of $a, c$. (a,b) $n_{\text{heads}} = 1$ (c,d) $n_{\text{heads}} = 4$. The accuracies for different $a$ and $c$ are exactly on top of each other.

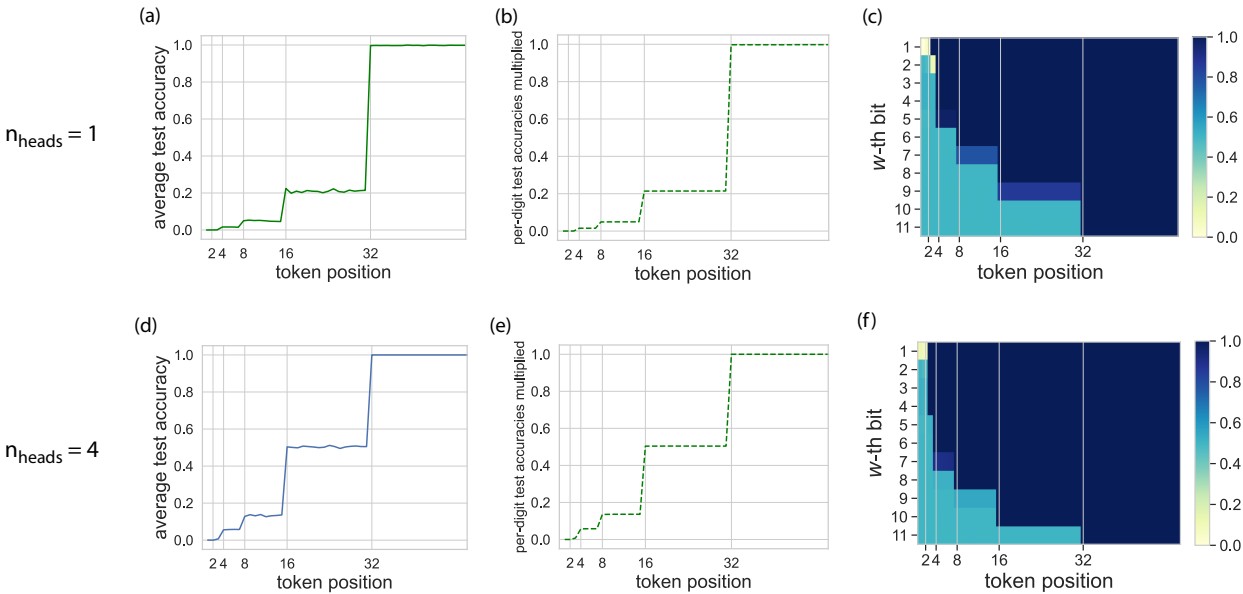

Figure 22. Test accuracy for $m = 2048 = 2^{11}$, depth=1, $n_{\text{heads}} \in \{1, 4\}$, $d_{\text{model}} = 768$. (a,d) Test accuracy averaged over $a, c$ and initial seeds. (b,e) Multiplication of per-digit test accuracies – matches exactly with the average test accuracy. (c,f) Per-digit accuracy in binary representation.

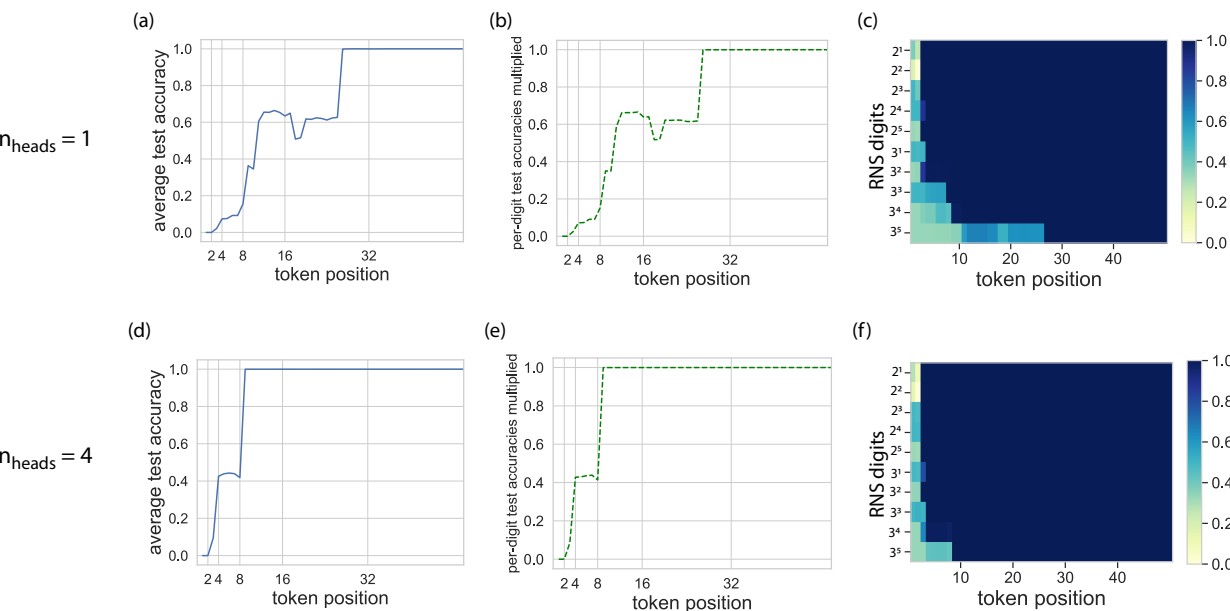

Figure 23. Test accuracy for $m = 7776 = 2^5 3^5$, depth=1, $n_{\text{heads}} \in \{1, 4\}$, $d_{\text{model}} = 768$. (a,d) Test accuracy averaged over $a, c$ and initial seeds. (b,e) Multiplication of per-digit test accuracies – matches exactly with the average test accuracy. (c,f) Per-digit accuracy in RNS representation.

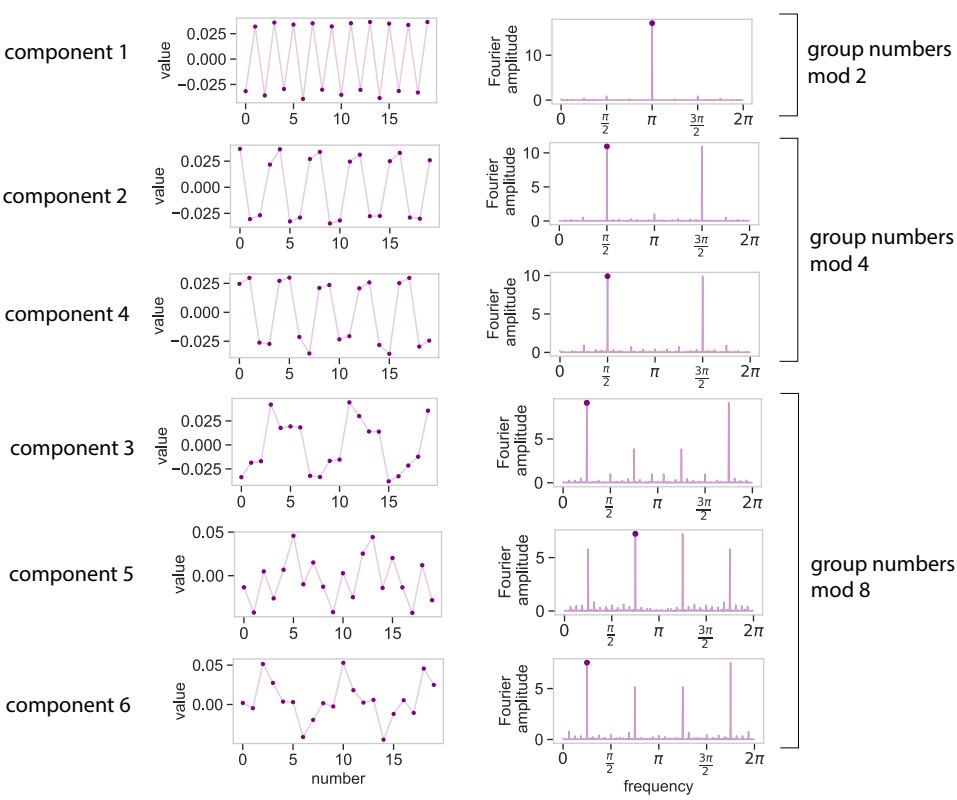

*Figure 24.* Projections along Top 6 principal components of the embedding matrix, for $m = 512$, depth=1, $n_{\mathrm{heads}} = 1$, $d_{\mathrm{model}} = 768$.

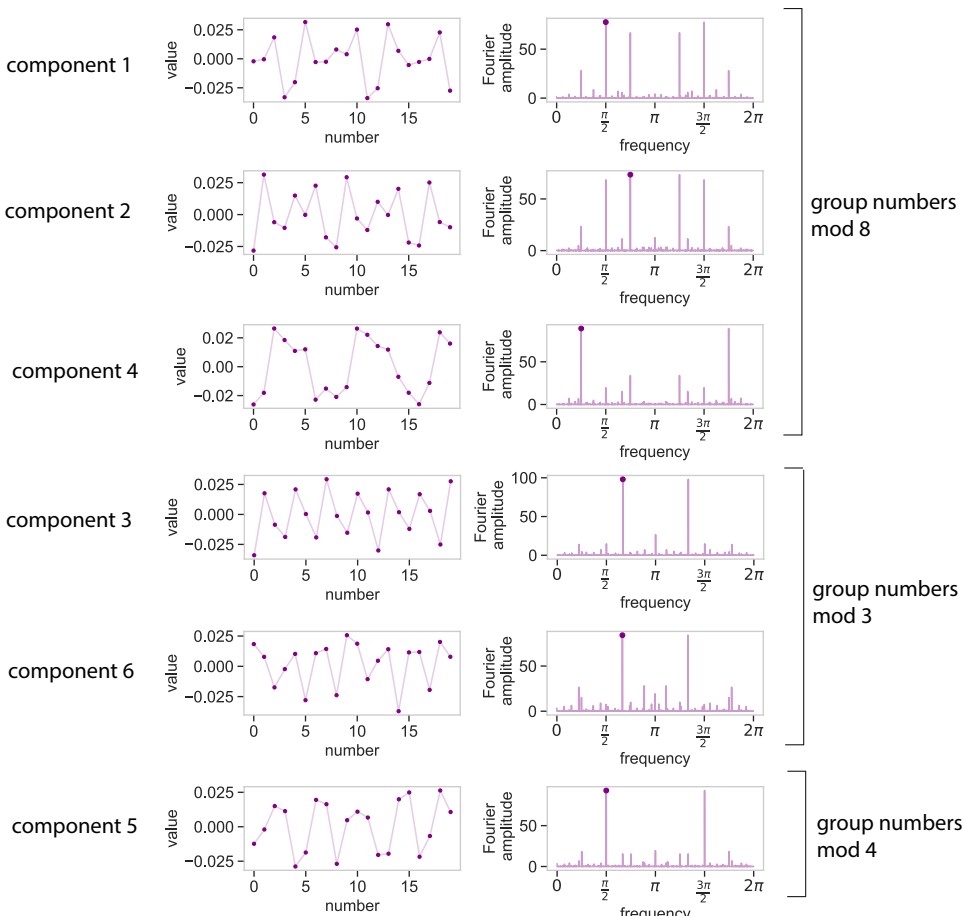

*Figure 25.* Projections along Top 6 principal components of the embedding matrix, for $m = 7776$, depth=1, $n_{\text{heads}} = 1, d_{\text{model}} = 768$.

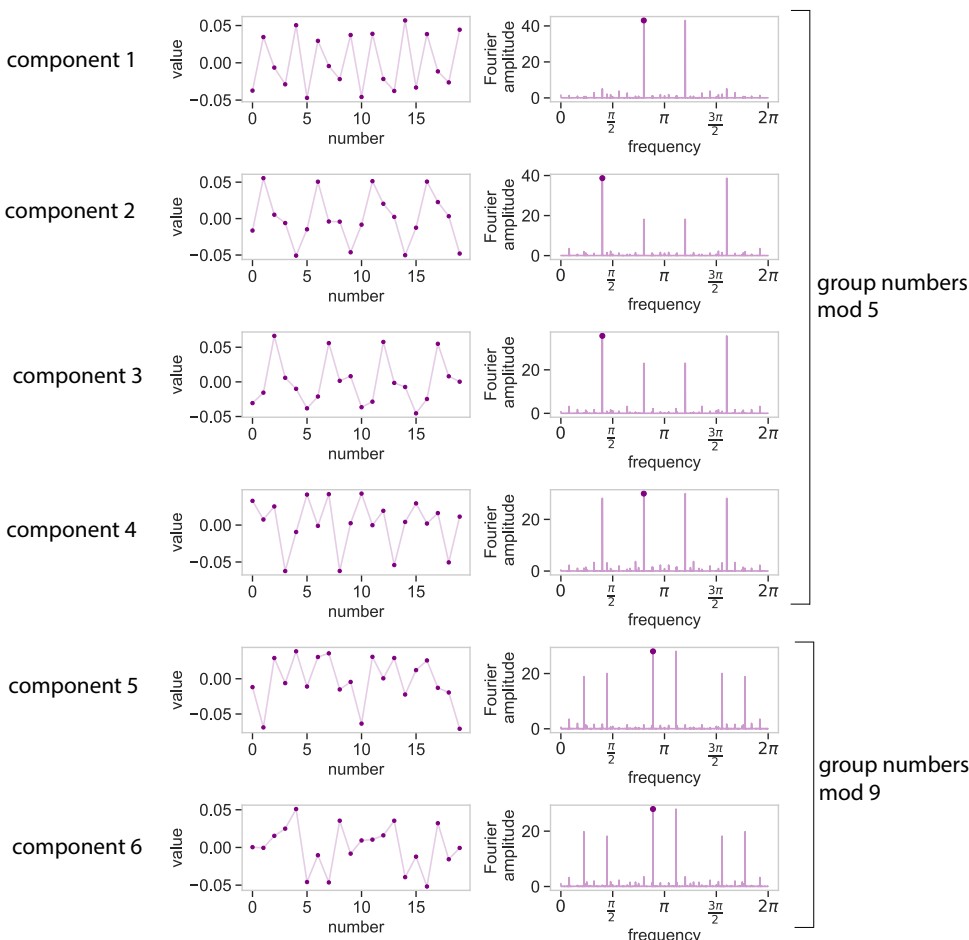

*Figure 26.* Projections along Top 6 principal components of the embedding matrix, for $m = 1800$, depth=1, $n_{\text{heads}} = 1$, $d_{\text{model}} = 768$.

# H. Unseen Modulus Interpretability

## H.1. Extra PCA plots for first layer heads

We present additional PCA analyses of attention heads shown in Figure 8, examining $\mathrm{PCA}(\boldsymbol{H}^{(h)}[:, t, :])$ for each head $h$ across various combinations of $a$, $c$, $x_0$, $m_{\text{test}}$, and position $t$. The results are depicted in Figure 27, where we include two additional first-layer heads (heads 2 and 3) that are responsible for performing operations related to modulo 5. Notably, with different choices of $a$, $c$, $m_{\text{test}}$ and $t$ compared to Figure 8 (a1, b1, c1), the clustering behavior remains unchanged.

Note the emergence of attention heads dedicated to processing modulo 2, 3, 5, and 7 can likely be attributed to the prevalence of these prime factors in the training set.

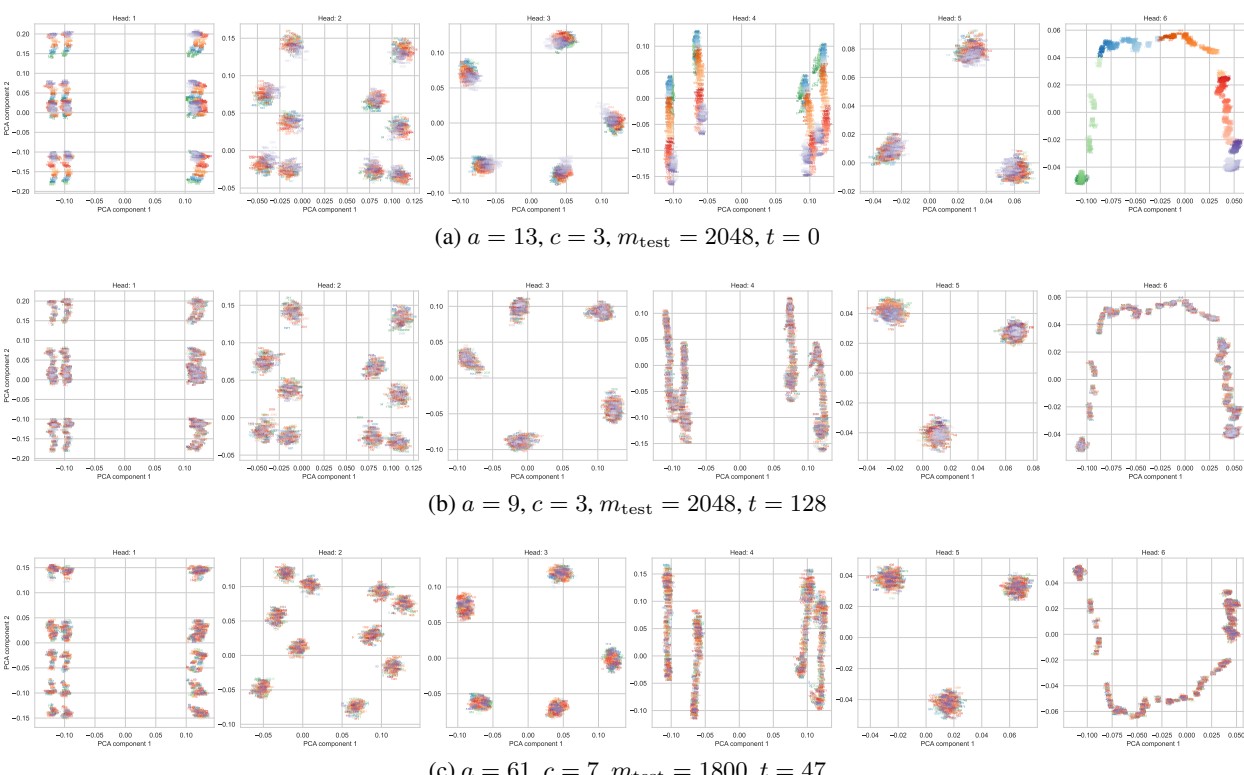

(a) $a = 13$, $c = 3$, $m_{\text{test}} = 2048$, $t = 0$

(b) $a = 9$, $c = 3$, $m_{\text{test}} = 2048$, $t = 128$

(c) $a = 61$, $c = 7$, $m_{\text{test}} = 1800$, $t = 47$

*Figure 27.* PCA analysis of first-layer heads. Although the ordering of numbers varies, the grouping behavior discussed in Section 4.3 remains invariant to changes in $a$, $c$, $m$, $x_0$, and $t$. Note that the last head is the head that appears in Figure 9, which does not exhibit a strong grouping bias.

## H.2. Pruning heads corresponding to irrelevant prime factors

To further validate our findings from Figure 8 in Section 4.3 regarding the correlation between attention heads and digit-wise accuracy, we conduct additional pruning experiments. The results of these experiments are presented in Figures 28 and 29.

## H.3. Patching other heads

In Figure 30, we present patching experiments for heads not previously shown in Figure 9. We focus only on selected heads where patching significantly affects predictions. Notably, none of these cases exhibit qualitative changes stemming from modulus alterations, further supporting our assertion that the head shown in Figure 9 is specifically responsible for estimating $m_{\text{test}}$.

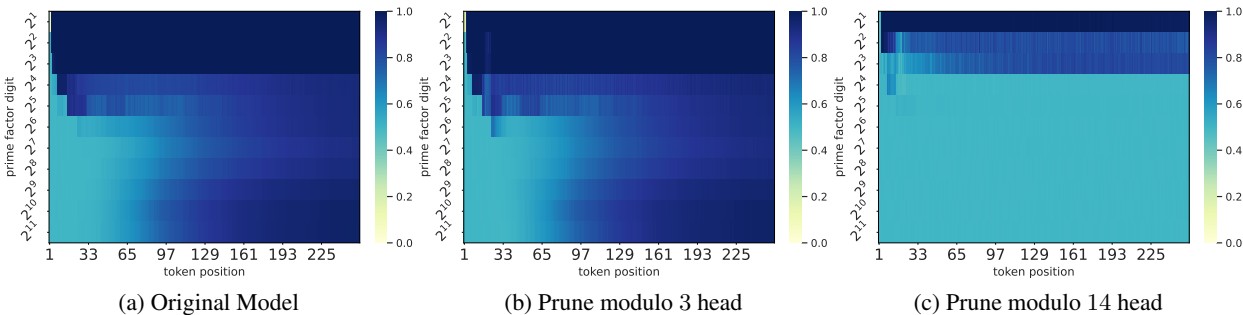

(a) Original Model        (b) Prune modulo 3 head        (c) Prune modulo 14 head

*Figure 28.* Per-digit accuracy for $m_{\text{test}} = 2048$ after pruning specific attention heads. (a) Results from the same model used in Section 4.3, identical to Figure 8 (b2); (b) Performance after pruning the attention head responsible for grouping numbers by their values modulo 3, which is irrelevant for solving sequences with $m_{\text{test}}$ (containing only the prime factor 2). After pruning, the model's performance shows marginal improvement for specific early bits at lower token positions; (c) Performance after pruning the attention head responsible for grouping numbers by their values modulo 14. The model's performance on $m_{\text{test}}$ decreases significantly, as this head partially contributes to processing relevant prime factors. However, since the primary head responsible for binary representation remains intact, the model maintains partial functionality.

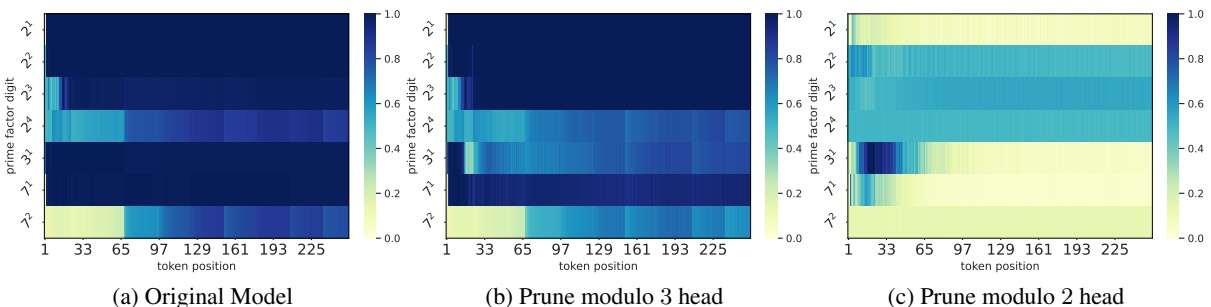

(a) Original Model        (b) Prune modulo 3 head        (c) Prune modulo 2 head

*Figure 29.* Per-digit accuracy for $m_{\text{test}} = 2352$ after pruning specific attention heads. (a) Results from the same model used in Section 4.3, identical to Figure 8 (b2); (b) Performance after pruning the attention head responsible for grouping numbers by their values modulo 3, which is relevant for one specific digit in this case. After pruning, the model's performance shows a clear degradation, with the strongest one happening exactly at the digit corresponding to modulo 3 (c) Performance after pruning the attention head responsible for grouping numbers by their values modulo 2. The model's performance on $m_{\text{test}}$ got obliterated, as there are many base-2 digits in the RNS representation of $m_{\text{test}}$ in this case.

### H.4. In-accurate estimation of $m_{\text{test}}$

From the cosine-similarity panel in Figure 9, we observe that the model's estimation approximates the target $m_{\text{test}} = 2048$. However, detailed analysis reveals that the highest cosine-similarity occurs at $m_{\text{est}} = 2033 = 19 \cdot 107$, with neighboring values exhibiting similarly high cosine-similarity values. If we assume $m_{\text{est}}$ represents the model's internal belief, then the prime representation would consist solely of powers of 19 and 107. Such a representation can only produce periodic structures for the $k$-th bit of a binary number when $r = \text{lcm}(19, 2^k)$. Consequently, patterns before reaching that period would appear random in this representation, providing a weaker signal compared to the correct representation. This explains why the model preferentially selects binary representation for lower bits when $m_{\text{test}} = 2048$.

For higher bits, the low-bit representation can be determined up to $2^k$ bits through copying. If the model utilizes this information in later stages, the precision of $m_{\text{est}}$ can be drastically improved. Specifically, when $\lfloor m_{\text{est}}/2^k \rfloor = \lfloor m_{\text{test}}/2^k \rfloor$, the model can identify the correct representation as long as $|m_{\text{test}} - m_{\text{est}}| < 2^k$. This argument can be extended to any other composite $m_{\text{test}}$. Note that this argument is hypothetical; further proof of this mechanism remains in future work.

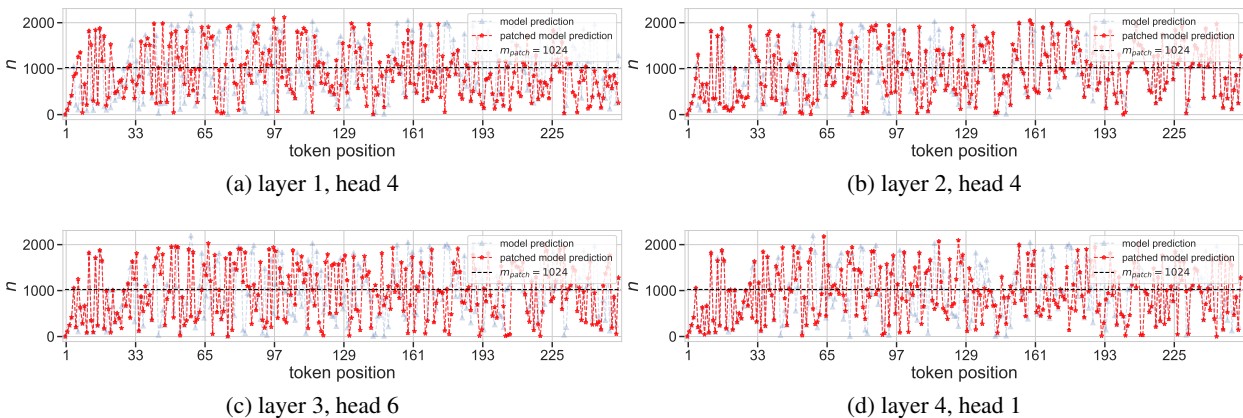

(a) layer 1, head 4

(b) layer 2, head 4

(c) layer 3, head 6

(d) layer 4, head 1

*Figure 30.* Patching experiments following the setting of Figure 9 in the main text. None of these heads, after patching, make the model believe that the modulus is close to $m_{\mathrm{patch}}$.

## H.5. Evidence for Step iii

In Figure 31, we present attention patterns and token distance statistics for a selected attention head in later layers, with the same model as the one used in Section 4.3. The token distances are measured by the spacing between keys with top-4 attention weights for a given query. Our analysis reveals that the model develops a head capable of dynamically adjusting their lookback distance for computations based on $m_{\mathrm{test}}$, which we interpret as evidence for **step iii** of the algorithm proposed in Section 4.3.

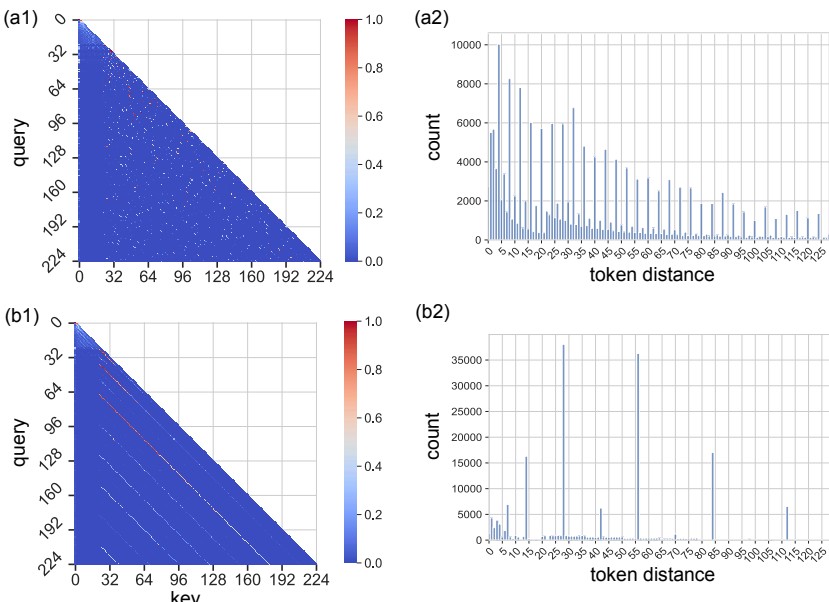

*Figure 31.* Attention patterns and token distance statistics for layer 2, head 3 of the model analyzed in Section 4.3. (a1, b1) Results for a sequence with $m_{\mathrm{test}} = 2048$, $a = 5$, and $c = 31$. The statistics reveal that the model consistently looks back at distances that are multiples of 4, which divides $m_{\mathrm{test}}$ and enables the correct copy behavior. (a2, b2) Results for a sequence with $m_{\mathrm{test}} = 2352$, $a = 85$, and $c = 5$. In contrast to panels (a1, b1), the same attention head now consistently looks back at distances that are multiples of 14, which allows the model to copy lower digits from the context. This adaptive behavior demonstrates that the model has acquired the ability to dynamically adjust its lookback distance by $r$ iterations to copy the lower digits and leave the higher digits to the later layers for computation.

# I. Scaling Up the Modulus

### I.1. Base-b tokenization

To convert an integer $x$ to base $b$ with the least significant digit (LSD) first, repeatedly divide $x$ by $b$, storing remainders:

$$d_k = x \mod b, \quad x = \lfloor x/b \rfloor. \tag{17}$$

Stop when $x = 0$. The sequence $(d_0, d_1, \dots)$ is the base-$b$ representation in LSD-first order.

Example: Converting $x = 3,214,748,365$ to base 256:

$$3,214,748,365 \div 256 = 12,557,610 \text{ remainder } 205, \quad d_0 = 205,$$
$$12,557,610 \div 256 = 49,053 \text{ remainder } 42, \quad d_1 = 42,$$
$$49,053 \div 256 = 191 \text{ remainder } 157, \quad d_2 = 157,$$
$$191 \div 256 = 0 \text{ remainder } 191, \quad d_3 = 191.$$

The final LSD-first representation consists of four tokens:

$$(205, 42, 157, 191)_{256}.$$

Each token is then one-hot encoded into a $b$-dimensional vector, where only the index corresponding to the token value is set to 1. These one-hot vectors are then fed into a token embedding layer with an embedding dimension of $d_{model} = 1024$.

### I.2. Abacas Embeddings

The positional embedding for the $j$-th lower digit of the $i$-th number in the sequence is defined as:

$$\text{PosEmbed}(T_{i,j}) = E_{\text{int}}(i) + E_{\text{digit}}(j) \tag{18}$$

where $T_{i,j}$ represents the token corresponding to the $j$-th digit of the $i$-th integer.

- $E_{\text{int}}(i), E_{\text{digit}}(j) \in \mathbb{R}^{d_{\text{model}}}$ are learnable embeddings.

- $E_{\text{int}}(i)$ encodes the integer's position in the sequence.

- $E_{\text{digit}}(j)$ encodes the digit's relative position within the integer.

This embedding scheme ensures that each token captures both the integer's global position and the byte's local position. Figure 32 provides a visualization of base-$b$ tokenization and abacus embedding.

### I.3. Fixed Modulus

For each modulus $m = 2^k$, where $k \in [16, 32]$, we train a 2-layer GPT model with an embedding dimension of 1024 and a vocabulary size 256. The train set consists of $n_a = 1024$ multipliers and $n_c = 1024$ increments, selected via the Hull-Dobell theorem. One LCG sequence of length 512 is included in the train set for each (a, c) pair, resulting in a total training set size of $n_a \times n_c = 1,048,576$. For each modulus, the model is trained for 200,000 steps with a batch size of 512. The context length is $512\times$ the digit length of $m$ in the byte representation $-1$. For $m = 2^{32}$, the digit length in byte representation is 4; therefore, the context length is 2047. Training was performed using 4 A100 GPUs over a total duration of 21.82 hours. For $m = 65536$, the digit length in byte representation is 2, resulting in a context length of 1023, with training taking 4.83 hours. For each modulus, the test set includes 512 unseen $a$ values and 64 unseen $c$ values selected via the Hull-Dobell theorem.

The model may converge to different solutions depending on two random seeds: one for model initialization and batch shuffling, and another for dataset generation. Figure 35 shows the median performance across five runs, with the shaded region representing the range between the minimum and maximum values. For larger moduli, not all models successfully find a solution that achieves $100\%$ test accuracy.

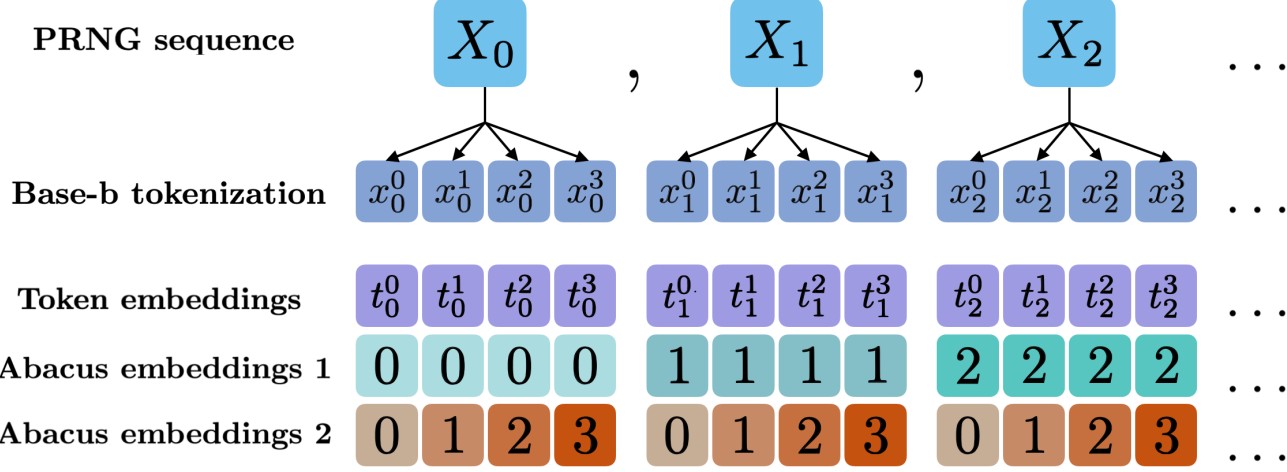

*Figure 32.* Visualization of base-$b$ tokenization and abacus embeddings. Abacus embedding 1 is shared by all the digits within the integer, while Abacus embedding 2 varies within the digit but is shared by all integers.

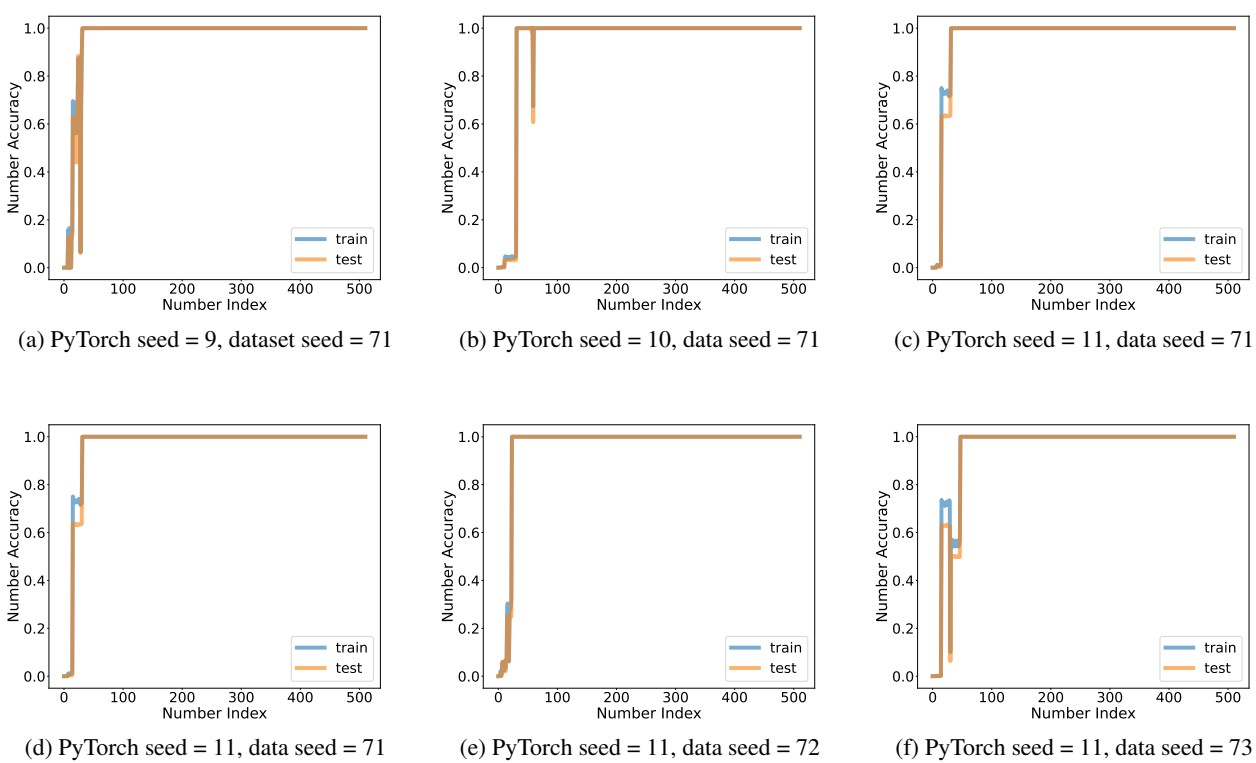

(a) PyTorch seed = 9, dataset seed = 71     (b) PyTorch seed = 10, data seed = 71     (c) PyTorch seed = 11, data seed = 71

(d) PyTorch seed = 11, data seed = 71     (e) PyTorch seed = 11, data seed = 72     (f) PyTorch seed = 11, data seed = 73

*Figure 33.* Test accuracy vs. Number index for $m = 2^{20}$. First Row: Three models trained on the same dataset, each using a unique PyTorch random seed that controls model initialization and batch shuffling. Second Row: Three models trained on different datasets, with each dataset generated using a unique random seed controlling NumPy randomness for sampling $a$, $c$, and $x_0$. All models converged to solutions that achieved and sustained 100% test accuracy, but differed in the number of in-context examples required to reach this performance.

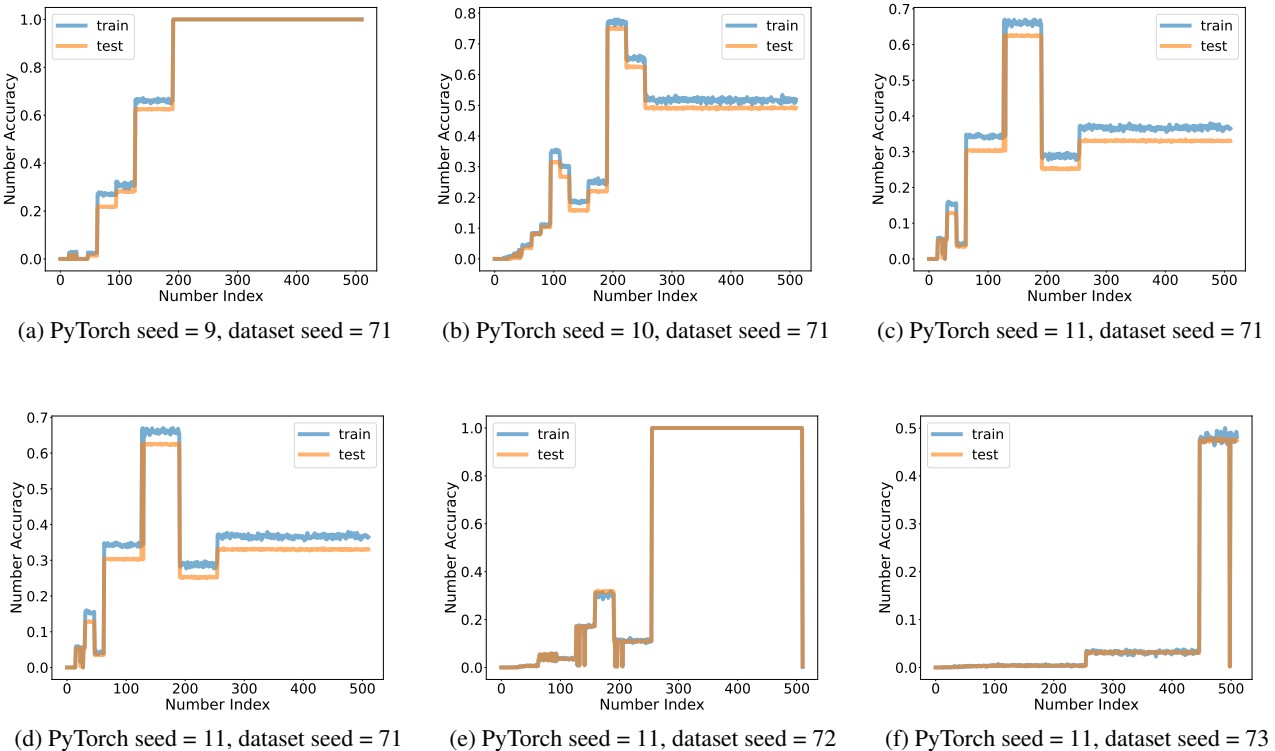

(a) PyTorch seed = 9, dataset seed = 71    (b) PyTorch seed = 10, dataset seed = 71    (c) PyTorch seed = 11, dataset seed = 71

(d) PyTorch seed = 11, dataset seed = 71    (e) PyTorch seed = 11, dataset seed = 72    (f) PyTorch seed = 11, dataset seed = 73

*Figure 34.* Test accuracy vs Number index for $m = 2^{32}$. First Row: Three models trained on the same dataset, each using a unique PyTorch random seed that controls model initialization and batch shuffling. Second Row: Three models trained on different datasets, with each generated using a unique random seed controlling NumPy randomness for sampling $a$, $c$, and $x_0$. Only one of the five models found a solution that achieved and sustained 100% test accuracy.

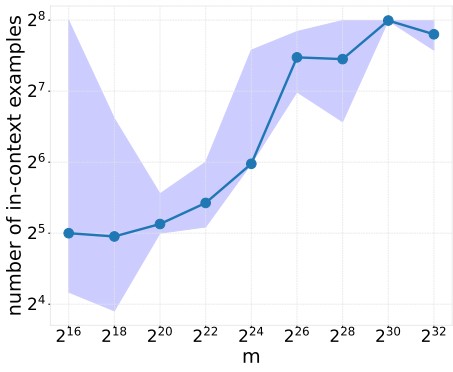

*Figure 35.* Median number of in-context sequence elements required to achieve 100% test accuracy across five runs. The shaded region represents the min-max range.

## I.4. Unseen Modulus

We train a 6-layer GPT model on a dataset that comprises $n_m = 32,768$ moduli, with $n_a = 128$ training $a$ values and $n_c = 1$ training $c$ values per modulus. This results in a total of $n_m \times n_a \times n_c = 4,194,304$ sequences, each of length 512 in the training set. In Figure 36a, where the tokenization base is 256, $1024 < m_{\text{train}} < 65536$. In Figure 36b, where the tokenization base is 243, $1024 < m_{\text{train}} < 59049$. Multipliers are selected based on the Hull-Dobell theorem when

sufficient qualifying $a$ values are available; otherwise, random $a$ values are used to ensure 128 multipliers for each modulus. The models with approximately 76M parameters were trained on 16 million sequences over 400,000 steps, using a batch size of 128 on a single H100 GPU for 22.62 hours. Because LCGs are typically defined for moduli that are powers of prime numbers, the model is tested on moduli that are powers of the primes 2, 3, 5, and 7. The test set consists of 512 unseen $a$ values and 64 unseen $c$ values selected via the Hull-Dobell theorem for each test modulus.

Test performance is influenced by the tokenization base, exhibiting a bias toward moduli that share the same base as the tokenization method. For instance, in Figure 36 (a) when using a byte-level representation, the model achieves better performance on moduli $m_{\text{test}} = 2^k$ compared to $m_{\text{test}} = 3^k$, $5^k$, or $7^k$. As contrasted with Figure 36 (b) where the tokenization base is $243 = 3^5$, the model performs better on moduli $m = 3^k$. This behavior is likely due to the property of LCGs, where for moduli that are powers of a prime $b$, the lowest $k$-th digit exhibits a period of $b^k$. Tokenization in such a base highlights this periodic structure, making it more apparent and easier for the model to leverage during training and prediction.

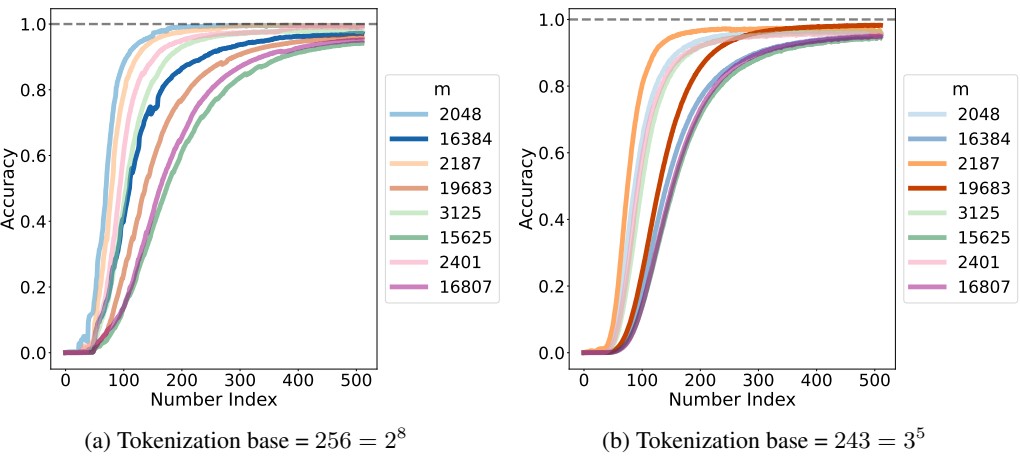

(a) Tokenization base $= 256 = 2^8$         (b) Tokenization base $= 243 = 3^5$

*Figure 36.* Test accuracy vs Number index. In (a), the moduli 2048 and 16384 (blue curves) have the same root 2 as the tokenization base 256. The model performs better on these two moduli. In (b), the moduli 2178 and 19683 (orange curves) have the same root 3 as the tokenization base 243. The model performs better on these two moduli.

