# OpenReview forum: "(How) Can Transformers Predict Pseudo-Random Numbers?"
_ICML.cc/2025/Conference — ICML 2025 poster_

### Official Review · Reviewer_tdrB · 2025-03-10

**Overall Recommendation:** 3

**Summary:**

The paper studies how Transformers can learn linear congruential generators (LCG), a class of simple pseudo-random generators. These are generators of the form $x_{n+1} = a x_n + c \mod m$ for some choices of $(a,c)$ and $m$. The authors demonstrate that Transformers can learn from data to simulate LCG in two settings: when the modulus is fixed, and when it is varied (which requires generalization to unseen values of $m$). The authors also analyze how the Transformer is able to compute the LCG, both when the modulus is fixed and when it is varied, and give a "pseudo-code" for the algorithm implemented by the Transformer. Additionally, the authors study how the depth and number of heads in the Transformer affect the performance on this task.

**Claims And Evidence:**

The results in the work are interesting, and to my knowledge the setting studied in this paper is novel. The target problem of generating pseudo-random numbers allows the authors to carefully study the algorithm learned by the Transformer in two settings. The paper is well-written and the authors do a good job in introducing their results. The algorithms that are learned by the Transformer are clearly introduced, and the authors show convincing evidence regarding the mechanisms learned by the Transformer model, demonstrating the the Transformer indeed learns to implement these algorithms.

The main concern that I have is regarding the motivation for the results discussed in the paper. What are the findings in the paper telling us about Transformers and/or PRNGs that we did not know before? The paper certainly demonstrates that Transformers are able to learn LCGs in novel settings, which is interesting, but I am not sure if this is particularly surprising given evidence on other similar problems like learning modular arithmetic (e.g., [1]). The algorithm that Transformers learn is certainly clever, but I am not sure why we should care about how Transformers solve this particular problem. Is this a new or interesting algorithm for computing LCG? Are the interpretability methods used in the paper novel compared to other, similar works? It seems to me that there has already been quite a few works on interpretability of Transformers on similar tasks (such as [1] and [2]), so clarifying exactly what are the novel contributions and conclusions of this work compared to previous works would be helpful.

Minor:
- A formal definition of the period $T_m(a,c)$ would be useful.
- In think there is a typo in the last paragraph on page 8 ($m2^16$ should be $m2^{16}$?)

[1] PROGRESS MEASURES FOR GROKKING VIA MECHANISTIC INTERPRETABILITY, ICLR 2023

[2] Feature emergence via margin maximization: case studies in algebraic tasks, ICLR 2024

**Essential References Not Discussed:**

See above.

**Experimental Designs Or Analyses:**

See above.

**Methods And Evaluation Criteria:**

See above.

**Other Comments Or Suggestions:**

N/A

**Other Strengths And Weaknesses:**

N/A

**Questions For Authors:**

See above.

**Relation To Broader Scientific Literature:**

See above.

**Theoretical Claims:**

See above.

---

> ### Author Rebuttal · Authors · 2025-04-01
>
> We thank the reviewer for their thoughtful assessment and candid feedback. To address the reviewer's main concern, we clarify the motivation behind and the contributions of our work.
>
> One major goal is to answer the following important question: To what extent can deep neural networks crack various primitives in cryptography? PRNGs are an important and commonly used component in cryptography, making them an ideal starting point for investigation. Moreover, transformers are perhaps the most effective pattern recognition systems ever developed, while PRNGs provide some of the best ways of hiding deterministic patterns, so it is natural to pit them against each other and study the resulting learning dynamics. Among PRNGs, LCGs represent the simplest case to examine, and we demonstrate how neural networks can successfully learn the subtle patterns in these sequences.
>
> This work is expected to be among the first in a series of studies examining neural networks' ability to learn increasingly complicated arithmetic sequences. This line of research may reveal completely unknown properties of widely used cryptographically secure PRNGs (CSPRNGs) like AES-CTR\_DRBG (also suggested by the Reviewer rtPM).
>
> Another important goal is to explore the learning ability of transformers in controlled settings, where the data generation process is fully understood, unlike for real-world data like natural images and language. PRNGs provide a natural setting to study in-context learning ability, sample efficiency, and the role of architectural complexity, and further add to discoveries in interpretability of neural networks.
>
> Next, we highlight our contributions:
>
> - **Unexpected scaling behavior**:
> We found how the number of in-context examples needed scales with problem complexity. One could argue that classical ways of breaking PRNGs exist in literature; however, we want to emphasize that our setup is qualitatively different because models are not explicitly informed that the data are LCG sequences. When training a Transformer model with just a collection of numbers without such context, there is no clear prior to determining whether the scaling law would be independent, sublinear, superlinear, or exhibit some other behavior. In other words, the scaling law pertains not merely to discovering $(a, c, m)$ or predicting the next number, but also to inferring the LCG rule.
>
> - **New interpretability results**:
>     The ability to spontaneously factorize numbers as part of the learned algorithm was new and surprising. This, along with the capacity to estimate moduli, adds to a short but growing list of results in mechanistic interpretability studies.
>
>     Specifically, although LCGs utilize modular operations, these tasks are drastically different from classification settings like [1] and [2] and have much stronger long context dependence than (He et al., 2024), where they stack examples from the same modulus in random order. In our setting, for the Fixed-Modulus (FM) case, this difference leads us to a completely different set of features, where the model converts numbers implicitly to the digit-wise (mixed radix or RNS) representation without forming any circular embeddings like those found in existing works. On the other hand, since the modulus keeps varying for the Unseen-Modulus (UM) case, the circular patterns observed in modular arithmetic cannot be used here. In this case, the model has to develop the ability to estimate the modulus and combine it with similar features as FM cases to solve the task. To summarize, despite the surface-level similarity to modular arithmetic, the model learned vastly different underlying algorithms.
>
>     Finally, our findings do not necessarily provide novel mathematical insights beyond what human experts in PRNG development already understand. However, the models' autonomous discovery of periodic structures in RNS representations from LCGs without explicit guidance is noteworthy. This particular phenomenon (please check the updated theoretical claim for Reviewer 7WLW) is likely familiar only to specialists in the field. The fact that this discovery occurred without directed instruction has not been previously documented in the literature and represents an exciting contribution. It suggests that sufficiently powerful models can potentially uncover unknown and surprising patterns from cryptographically secure PRNGs that humans are currently unaware of.

---

### Official Review · Reviewer_EiTT · 2025-03-14

**Overall Recommendation:** 4

**Summary:**

This study analyzes whether a Transformer based on next-token prediction can learn an LCG sequence and, if so, how it models the sequence. Specifically, the study demonstrates that a Transformer can learn an LCG sequence given sufficient architectural capacity and training data. Subsequently, it examines the algorithms learned by the Transformer in both the FM and UM cases. Additionally, the study explores the training recipe for scaling up the modulus in this task.

**Claims And Evidence:**

As shown in Figure 1, it is evident that the Transformer can successfully perform this task. Furthermore, based on the characteristics of the LCG sequence, the analysis of the algorithm learned by the Transformer is also clear.

**Essential References Not Discussed:**

I believe there is no missing essential work.

**Experimental Designs Or Analyses:**

The experimental design of this study is well-balanced and valid, as it addresses LCG sequence learning across various scenarios, avoiding biases in the experimental setup.

**Methods And Evaluation Criteria:**

Since this study does not specifically propose a new method, there is not much to cover in this section. However, one point of interest is the use of Abacus Embeddings in the modulus scaling process. It would be beneficial to include an analysis of this aspect.

**Other Comments Or Suggestions:**

There seem to be some minor typos in the Discussion section on page 8 that need to be corrected.

**Other Strengths And Weaknesses:**

Strengths: Selecting an appropriate task to analyze the capabilities of a Transformer is highly meaningful. While some previous studies have proposed interesting tasks, they often involved somewhat contrived scenarios. However, the task explored in this study deals with a random number generator used in actual cryptographic applications, making it a more thought-provoking and relevant choice.

Minor Weakness: It would be beneficial to discuss how the Transformer analysis conducted in this study could be applied to more general tasks, such as those in the NLP domain.

**Questions For Authors:**

If the study employs an autoregressive Transformer in the style of GPT, does that mean it uses a decoder-only Transformer operating with causal attention? I am curious whether causal attention is critically related to the study's findings. Additionally, I wonder whether an encoder-decoder or a prefix decoder-only Transformer architecture could also learn the LCG sequence, or perhaps even perform better.

**Relation To Broader Scientific Literature:**

The analysis of how the Transformer learns LCG is a valuable finding for future Transformer architecture design. Additionally, the task proposed in this study can serve as a foundation for further research, encouraging continued exploration of the inner workings of Transformers.

**Theoretical Claims:**

I have reviewed the theoretical claims and have no points to discuss regarding this aspect.

---

> ### Author Rebuttal · Authors · 2025-04-01
>
> We thank the reviewer for their encouraging comments and thoughtful questions.
>
>
> - **Questions**: Yes, our study employs a decoder-only Transformer architecture with causal masking (autoregressive).
>
>   This choice emulates the real-world scenario where observations are obtained sequentially and, at the same time, prevents the model from cheating by checking future information.
>
>   We agree that exploring other architectures would certainly be interesting. While a prefix decoder-only model might allow us to use a sequence design similar to our current approach, an encoder-decoder architecture would likely require changing our objective (for data efficiency), formatting the task as a translation from the sequence $x_0, x_1, \cdots, x_n$ to the parameters $(a, c, m)$. This reformulation would likely involve different learned features and scaling laws, and we leave them for future work.
>
> - **Relation to General Tasks**: This is certainly an interesting comment. As we mentioned in the introduction, LCGs can be viewed as a special kind of formal language, the latter of which is deeply intertwined with NLP studies. We believe that starting with LCGs and incrementally increasing the complexity of our datasets would eventually allow us to build a systematic approach to studying formal languages and NLP tasks.

---

### Official Review · Reviewer_7WLW · 2025-03-16

**Overall Recommendation:** 3

**Summary:**

This paper studies how Transformer can learn linear congruential generators sequence with either fixed or changing modulus. For fixed module, they discover that Transformer learns the radix representation and predicts each radix digit almost independently. Notably, the lower digit are predicted by copying from previous period. However, the model is also capable of predicting non-repeating higher order bit. For changing module, they discovered that the model learns to cluster number based on modulus for small numbers (like 2,3,4, 6) in embedding and first-layer attention head. They also find an attention head that attend to the largest number in the current sequence and use this to estimate the modulus.

**Claims And Evidence:**

Overall, the mechanism pointed out by the authors are well supported by both observation and intervention. However, some of the mechanism remains mostly unexplained:

1. In the fixed module case, the model is capable of predicting non-repeating bit in the modulus 2048 case, this is further shown in the scaling experiment that model only need example with m^{1/4} length to make successful prediction. How are these bit predicted by the model is not explained by the authors (while the authors point out that attention head that attend to 2^{k'} number ago is useful for this prediction.

2.  In the changing module case, the authors argue that the mechanism learned by later layer is 'similar' to the unseen module case, but they only showcase that the attention head in higher layer will similar look back over period in the appendix. This evidence is not sufficient enough to support such a claim.

**Essential References Not Discussed:**

I think this paper needs to discuss relationship with in-context linear regression (with LCGs as regression in the modulus setting) and related work in using Transformer to solve LWE (regression in modulus setting with noise). Some of the modulus calculation interpretability work is also missed. An incomplete list includes [1, 2, 3].


[1] https://arxiv.org/abs/2211.15661
[2] https://arxiv.org/abs/2306.17844
[3] https://arxiv.org/abs/2207.04785

**Experimental Designs Or Analyses:**

The design and analysis is overall detailed and complete. I have raised my concern in the Claims and Evidence part.

**Methods And Evaluation Criteria:**

The paper train and interpret different models with both fixed and changing modulus. The evaluation is complete and thorough.

**Other Comments Or Suggestions:**

N/A

**Other Strengths And Weaknesses:**

The scaling experiment is very original and interesting.

**Questions For Authors:**

1. How is the mixed radius experiment carried out in detail? Especially, how is the mixed radius calculated for the experiment here?

2. What is the possible mechanism that enables prediction beyond calculating?

3. What implication will this work bring for the PRG community? Is LCG with known modulus actually easier to break with optimal breaking algorithm than the scaling law shown here (as it only requires regressing two numbers)? How is Transformers compared with the current best breaker in terms of complexity in the Unknown modulus setting?

**Relation To Broader Scientific Literature:**

This paper is similar to other interpretability work that studies how Transformer performs modulus calculus. One difference is that this paper does not discuss how the modulus calculus is calculated here.

**Theoretical Claims:**

There is some theoretical flaws in the main paper about mixed radix setting.

Equation (2) has a typo, there should be a summation from 0.

Equation (3) are incorrect in general, one can easily see this as some small number can easily have two different mixed radius representation under this setting ( 1 * 2 + 1 * 2^2 = 2 * 3). Some of the number are not represented by this representation. This makes the result beyond power of 2 in fixed modulus case hard to understand.

---

> ### Author Rebuttal · Authors · 2025-04-01
>
> We thank the reviewer for giving a detailed feedback and raising important questions. **Note**: New experiment figures at: https://doi.org/10.6084/m9.figshare.28703570.v2
>
> ## Theoretical claims and related experiments
> The reviewer is correct in pointing out that equation (3) is incorrect in general and incompatible with the experiments in Figures 3 and 6 (i.e. our experiments are accurate, but equation (3) is in error). Below, we present the corrected version of equation (3) and surrounding text:
>
> We used the Residual Number System (RNS) (Garner 1959), where each number is represented by its values modulo pairwise coprime factorizations of $m$. Specifically, consider sequences with a composite modulus $m$, which has a prime factorization $m = p_1^{w_1} p_2^{w_2}\cdots p_q^{w_q}$. In this case, we can *uniquely* represent each number $(x \text{ mod } m)$ as the tuple of residuals $( x \text{ mod } p_1^{w_1}, x \text{ mod } p_2^{w_2}, \dots, x \text{ mod } p_q^{w_q} )$ and similar to equation (2) we can further decompose each residual,
> $$x \text{ mod } p_j^{w_q} = \alpha_{j,0} p_j^0 + \alpha_{j,1} p_j^1 + \cdots + \alpha_{j,w_j-1} p_j^{w_j-1}$$
> where $\alpha_{j,w} \in \{ 0, 1, \dots, p_j - 1 \}$ are base-$p_j$ digits. We refer to $\{\alpha_{j,w}\}$ as the "RNS representation". When period $\mathcal T_m = m$, we can show that each digit $\alpha_{j,w}$ has a period of $p_j^w$. Then, the rest of the discussion remains the same. For example, the $r$ step iteration still reduces the period of each digit $\alpha_{j,w}$ from $p_j^w$ to $p_j^w / \gcd(r, p_j^w)$.
>
> In experiments for Figures 3 and 6, we calculated the collection of $\{\alpha_{j, w}\}$ for the target and model predictions, then compared which $\alpha_{j,w}$ matching. Since these experiments already implicitly use RNS representations, they remain unchanged.
> (We had erroneously thought that the mixed-radix representations are similar to RNS representations -- however, we later realized that this is not the case.)
>
> ## UM is similar to FM
>
> The key difference between FM and UM is that the model has to figure out the modulus in-context. Once the model determines $m$, the sequence prediction task for UM is not any different from the FM case. Beyond the attention heads we show in the appendix, the per-digit accuracy in Figure 6 also shows a ladder-like structure similar to FM cases, suggesting the copying nature of the UM case.
>
> Such $r$-step copying behavior is crucial for the model to make correct predictions even when the estimate for the modulus is imprecise. The model achieves this in steps: i) it estimates the modulus $\widetilde m_{\mathrm{est}}$ using the largest number observed in the context so far -- which is usually slightly smaller than the actual modulus $\widetilde m_{test} < m_{test}$. ii) The model then corrects this imprecision by leveraging the $r$-step copying behavior. This effectively reduces the task to modulo $m / r$. Then, the small difference in estimating the modulus gets rounded off like $m/r = \lfloor m_{\mathrm{estimate}} / r \rceil$.
>
>
> ## Mechanism for predicting higher bits
> Due to space constraints, we refer the reviewer to the "Learning of higher bits" section of our response to Reviewer rtPM.
>
>
> ## Related works
> We thank the reviewer for pointing out relevant references. We will cite these references and add a thorough discussion on similar works in the final version of the paper.
>
> ## Questions
> We have addressed Questions 1&2 above.
>
> Q3. To the best of our knowledge, all LCG breakers *assume that the underlying sequence is LCG* and focus on estimating the parameters $(m, a, c)$ [1] or predicting the next number [2]. The strong prior on the form of the sequence makes the task much simpler. (termed "open-box" in cryptography).
>
> In contrast, the Transformer has no such prior, thereby making the task significantly harder than simply finding $(m, a, c)$ (termed "closed-box"). During training, it learns to parse and utilize the patterns in sequences of seemingly random numbers, and applies these abilities for out-of-distribution sequences at inference time. Thus, it is difficult to make a fair comparison between the "closed-box" Transformer approach with "open-box" optimal breaking algorithms.
>
>  We emphasize that the primary purpose of this paper is not to break LCGs with optimal sample complexity, nor to compete with existing open-box algorithms. Rather, our main objective is to investigate the extent to which a Transformer model can break various cryptographic primitives -- this paper serves as an initial step towards that goal. That the model can identify patterns already known to experts is encouraging, suggesting that Transformer models might also discover previously unknown patterns in more complex PRNGs (see also our response to Reviewer tdrB).
>
> [1] J. B. Plunstead, Inferring a Sequence Generated by a Linear Congruence, 1982
>
> [2] J.Stern, Secret linear congruential generators are not cryptographically secure, 1987

---

### Official Review · Reviewer_rtPM · 2025-03-17

**Overall Recommendation:** 4

**Summary:**

Summary:
This paper trains transformers on the task of in-context predicting the next element of a sequence generated with a Linear Congruential Generator (LCG). An LCG has the form:
x_{n+1} = a x_n + b (mod m), where
a, b, m are unknown numbers.

The paper studies two settings: 1) Fixed-Modulus (FM) setting, where m is the same in all contexts, and 2) Unseen-Modulus (UM) setting, where m varies, and the model is tested on held-out values of m.

The paper mechanistically interprets transformers on these two tasks. It argues that the radix representation for (mod m) is critical to how the model internally learns to represent the input. It argues that the embedding of the inputs reflects their structure in the radix representation.

Additionally, it argues that the model learns the lower-order bits by performing a lookup to previous terms in the sequence whose period corresponds to the radix representation.

**Claims And Evidence:**

The experiments were convincing to me -- with ablation and patching studies that confirmed the researchers' interpretations of the internal workings of the model.

1) However, I do not understand how the higher-order bits in mod 2048 sequences, are found by the model. This was mysterious to me, and I felt like it was not addressed by the paper. The paper mentions that if you look at the sequence x_{i}, x_{i+r}, x_{i+2r} for some r dividing 2048, then the higher-order bits have a period of length 2048 / r. However, the model does not have access to enough terms in this sequence to perform a backwards lookup in the same way that it can for the lower-order bits.

So what is the mechanism to learn these higher-order bits? I couldn't find the answer to this question in the paper, although the paper mentions "multiple faint lines" in the attention scores "that are 2^{k'} distance apart" for k' < k-1, and that it "combines information from all these tokens to predict the higher bits". This felt too vague for a paper that has as an objective to fully mechanistically interpret the network.


2) Another related question that I have is: do you find that it is much harder to learn in the fixed-modulus setting when m is a prime? In that case the trick that you have described for learning lower-order digits no longer applies. When the model groks, what is the mechanism by which it learns the LCG in that case?

**Essential References Not Discussed:**

Not that I am aware of

**Experimental Designs Or Analyses:**

Yes, I looked through the entire paper and it seemed OK. See my comments on "claims and evidence" above, since I think that a full mechanistic understanding is not obtained by this version of the paper.

**Methods And Evaluation Criteria:**

Yes

**Other Comments Or Suggestions:**

* Typo lines 105-109. The variable $m$ should be $p$ instead.
* Typo Line 167? Equation (3) should be a product of the sums, instead of a sum of the sums ?
* Typo in Figure 10 caption. eample should be example.
* Typo in Line 436, column 2. 2^16 should be 2^{16}.
---
* The answer in https://security.stackexchange.com/questions/4268/cracking-a-linear-congruential-generator shows that LCGs can be cracked with a simple classical algorithm -- they have notably bad security. It is not as much of a surprise, therefore, that transformers can learn to crack LCGs than if they could crack more complicated schemes. Do you think that you could train a transformer to crack encryption schemes like AES? The scheme does not have strong theoretical backing insofar as I am aware, so it could be an excellent candidate to try.

**Other Strengths And Weaknesses:**

I think that the setting is original and the results are cleanly described and presented. With the exception of the explanation of the higher-order bits, I think that this paper does a good job understanding how neural nets learn LCGs.

Thus, I think of this paper as resolving the case where m is a product of a few small primes: e.g. m = 2310 = 2*3*5*7*11 seems satisfactorily addressed by this paper, but for m = 2048 I am not fully convinced by the paper yet.

**Questions For Authors:**

Could you please clarify my confusions about higher-order bits above -- either by pointing to what I am missing in the current paper or by providing a new analysis that would convince me? I would raise my score.

**Relation To Broader Scientific Literature:**

This paper fits into work on mechanistic interpretability, which seeks to reverse-engineer trained neural networks. I am familiar with the results on neural networks learning arithmetic, and there are some elements in this work that vaguely resemble that. However, I am not aware of papers that do what this paper is doing, so it seems original to me.

**Theoretical Claims:**

No theoretical claims. I checked the Hull-Dobell theorem and they have indeed cited it right.

---

> ### Author Rebuttal · Authors · 2025-04-01
>
> We thank the reviewer for their careful assessment and valuable feedback. We have added new experiment Figures S1, S2 at the link: https://doi.org/10.6084/m9.figshare.28703570.v2 -- which we will refer to in our response below.
>
> ## Learning of higher bits
>
> Below we present a modified and more accurate account of copying and predicting behavior in the FM setting. The corrections and modified experiments mentioned here will be reflected in the final version of our paper.
>
> Consider the model predicting the number $x_t$. For copying the last $k$ bits (least significant), attending to only one token is required -- located at position $t - 2^k$, where $k = \lfloor \log_2 t \rfloor$. This can be seen from our new experiments in Fig.S1(a). Note that this is a stronger statement than the one in the paper, where we mentioned the need for two bright lines for copying.
>
> Attention to all the other tokens contributes towards predicting higher bits. Specifically the attention to token position $t - 2^{k-1}$ plays an important role. Note that this is a correction to the comment in the paper that the line at $t - 2^{k-1}$ only contributes to copying. This is demonstrated in Fig.S2(b), which is an improved version of Fig.15. (We found a subtle mismatch between selecting the attention mask using top two values and using desired two token positions.)
> It shows that even if we mask out attention to all tokens except positions $t-2^k$ and $t - 2^{k-1}$, the model can predict higher bits with remarkable accuracy. Here we present an intuitive explanation of predicting the higher bits via a simple example, using the sequence from Fig.2 in the paper: $(1962, 411, 624, 1593, 838, \dots)$.
>
> Consider the model predicting the number at $t=4$ ($k=2$). In predicting $x_4 = 838 \equiv (01101000110)_2$, the model attends to $x_0 = 1962 \equiv (11110101010)_2$ and $x_2 = 624 \equiv (01001110000)_2$.
> We label the numbers by their lowest 3 bits: $x_0 \equiv (\cdots 010)_2$ and $x_2 \equiv (\cdots 000)_2$. Then, by copying the lowest two bits, we have $x_4 \equiv (\cdots ?10)_2$. Now, if we consider $r=2$ iterations and drop the (constant) lowest bit, we obtain the new reduced sequence $x'_0=x_0 \equiv (\cdots 01)_2$, $x'_1=x_2 \equiv (\cdots 00)_2$, and $x'_2=x_4 \equiv (\cdots ?1)_2$. Since the second lowest bit of this reduced sequence has period $8/2=4$, the only possible way to satisfy the period condition for the last two digits in the reduced sequence $x'_t$ is to have $x'_2 \equiv (\cdots 11)_2$ (and $x'_3=x_6 \equiv (\cdots 10)_2$).
>
> In this way, the model calculated higher bits just by using the constraints from the period, along with the knowledge of $x_{t - 2^k}$ and $x_{t - 2^{k-1}}$.
> This argument can be extended to even higher bits, by considering $r=2^k$-step iterations and added constraints from digit-wise periods. In practice, this method can be made even more robust with the knowledge of "other faint lines".
>
> ## LCG with prime $m$
> We find that in the FM setting with prime $m$ the task becomes much harder. Since there are no digit-wise periodic patterns, the model cannot perform the algorithm described in Section 4.2. In Fig.S2, we trained two identical models to learn $m=2039$ and $m=2048$, and observed that $m=2039$ cannot be learned within the same number of training steps. To rule out potential constraints from model capability, we used depth 2 models (compared to depth 1 in the paper).
>
> Note that moduli made up of powers of small primes (e.g. $2^w$) are ubiquitous in practice due to their computational efficiency. This motivated our choice of moduli in the paper.
>
> ## Classical algorithms vs Transformers
>
> While LCGs can be cracked using classical algorithms, Transformers face a fundamentally different challenge in our setup -- they have to determine the underlying sequence in an LCG without explicit instruction. In addition, these Transformers have to reverse engineer an algorithm from seemingly random sequences. We think that without explicit training, it is non-trivial to predict if Transformers can learn to predict even these simple LCGs.
>
> Regarding AES and more sophisticated encryption schemes, we agree that these would be interesting targets for future research. Our work on LCGs serves as the first crucial step for predicting these complex encryption schemes using Transformers.
>
> For further discussion, we refer the reviewer to the "Unexpected scaling behavior" paragraph in our response to Reviewer tdrB.
>
> ## Corrected Equation 3
>
> Due to space constraints, we refer the reviewer to the "Theoretical claims and related experiments" section of our response to the Reviewer 7WLW.
>
> We hope that we have satisfactorily addressed all of the reviewer's questions, especially concerning learning of higher bits.

---

> > ### Comment · Reviewer_rtPM · 2025-04-06
> >
> > Reposted as a rebuttal comment because I realized authors cannot see official comments:
> >
> > Thank you for your detailed reply. The experiment in S1 is convincing me that your explanation is along the right lines (there is a typo in caption of S1 fyi with t-2^k). However, there is something I’m still missing. In your example above in the reduced sequence why can’t x_2’ end with 10 and x_3’ end with 11?
> >
> > Why is it necessary that x_2’ ends with 11 and x_3’ with 10? Is this just a fact about possible LCG sequences?
> >
> >
> > EDIT:
> > ------
> > Thank you for your response below, I think this clarifies things for me. If you could include an explanation of this in your paper (and the more general case for high-order bits of m = 2^k), I think that it would help readers a lot because the high-order bits were the most mysterious part of this analysis for me.

---

> > > ### Author Response · Authors · 2025-04-07
> > >
> > > We thank the reviewer for carefully following our rebuttal and engaging in follow-up discussion.
> > >
> > > The necessity that $x_2'$ has to end with $11$ and $x_3'$ ends with $10$ is indeed related to possible LCG sequences. Specifically, it follows from the digit-wise periods along the seuqence. We had left this fact implicit due to space constraints -- but we elaborate it here.
> > >
> > > Since we select the test sequences using the Hull-Dobell theorem, the original LCG sequence has period $m=2048=2^{11}$. In this case, $w^{th}$ lowest digit (bit) has a period of $2^w$ along the sequence. After copying the last digit, the reduced sequences ($r=2$) also maintains this property.
> > >
> > > Thus, in the redcued sequence $\\{x_0', x_1', x_3', x_4', \dots \\}$ the lowest digit has period 2: $\\{ (\cdots 1), (\cdots 0), (\cdots 1), (\cdots 0), \dots \\} $.
> > > Similarly, the second lowest digit has period 4: $\\{ (\cdots 0 \cdot), (\cdots 0 \cdot), (\cdots 1 \cdot), (\cdots 1 \cdot), \dots \\} $.
> > >
> > > Now if $x_2'$ ends in $10$ and $x_3'$ ends in $11$ then the resulting sequence $\{x_0', x_1', x_3', x_4', \dots \}$ would look like $\\{(\cdots 01), (\cdots 00), (\cdots 10), (\cdots 11)\\}$, which violates the digit-wise period property of the LCG sequence. Consequently, $\\{(\cdots 01), (\cdots 00), (\cdots 11), (\cdots 10)\\}$ is the only viable option.
> > >
> > > We are happy to provide further clarifications and answer any other questions the reviewer may have.

---

### Decision · Program_Chairs · 2025-05-01

**Decision:**

Accept (poster)

**Comment:**

This paper studies the high-level question of "To what extent can deep neural networks crack various primitives in cryptography?" In particular, this paper trains transformers on the task of in-context predicting the next element of a specific PRNG, ie, Linear Congruential Generator (LCG) and mechanistically interprets transformers.

On the positive side, the experiment design and analysis are overall solid and complete. The reviewers are convinced by the mechanism learned by the transformer revealed in the paper. All reviewers recommend to accept this paper and I think this paper is a good and solid contribution to ICML.

On the negative side, though the experiment design and findings are novel, the reviewers have concerons about the motivation of this work --- it is not crisply clear what the takeaway is from the paper, since LCG can be broken easily by classical algorithms. It is not too surprising that transformers can compute the same function which are easy to compute in classical computation models. The authors should include the related discussions in rebuttal into the next revision of the paper.